



# Assessment of the quality of TROPOMI high-spatial-resolution NO$_2$ data products

Xiaoyi Zhao[1], Debora Griffin[1], Vitali Fioletov[1], Chris McLinden[1], Alexander Cede[2,3], Martin Tiefengraber[3,4], Moritz Müller[3,4], Kristof Bognar[5], Kimberly Strong[5], Folkert Boersma[6,7], Henk Eskes[6], Jonathan Davies[1], Akira Ogyu[1], Sum Chi Lee[1]

[1]Air Quality Research Division, Environment and Climate Change Canada, Toronto, M3H 5T4, Canada.
[2]NASA Goddard Space Flight Center, Greenbelt, MD 20771, USA.
[3]LuftBlick, Kreith 39A, 6162 Mutter, Austria.
[4]Department of Atmospheric and Cryospheric Sciences, University of Innsbruck, Innsbruck, Austria.
[5]Department of Physics, University of Toronto, Toronto, ON, M5S 1A7, Canada.
[6]Royal Netherlands Meteorological Institute (KNMI), De Bilt, the Netherlands.
[7]Environmental Sciences Group, Wageningen University, Wageningen, the Netherlands.

*Correspondence to*: Xiaoyi Zhao (xiaoyi.zhao@canada.ca)

**Abstract.** The TROPOspheric Monitoring Instrument (TROPOMI) on-board the Sentinel-5 Precursor satellite (launched on 13 October 2017) is a nadir-viewing spectrometer measuring reflected sunlight in the ultraviolet, visible, near-infrared, and shortwave infrared spectral ranges. The measured spectra are used to retrieve total columns of trace gases, including nitrogen dioxide (NO$_2$). For ground validation of these satellite measurements, Pandora spectrometers, which retrieve high-quality NO$_2$ total columns via direct-sun measurements, are widely used. In this study, Pandora NO$_2$ measurements made at three sites located in or north of the Greater Toronto Area (GTA) are used to evaluate the TROPOMI NO$_2$ data products, including standard Royal Netherlands Meteorological Institute (KNMI) tropospheric and stratospheric NO$_2$ data product and a TROPOMI research data product developed by Environment and Climate Change Canada (ECCC) using a high-resolution regional air quality forecast model (used in the air mass factor calculation). It is found that these current TROPOMI tropospheric NO$_2$ data products (standard and ECCC) met the TROPOMI design bias requirement. Using the statistical uncertainty estimation method, the estimated TROPOMI upper limit precision falls below the design requirement at a rural site, but above in the other two urban and suburban sites. The Pandora instruments are found to have sufficient precision to perform TROPOMI validation work. In addition to the traditional satellite validation method (i.e., pairing ground-based measurements with satellite measurements closest in time and space), we analyzed TROPOMI pixels located upwind and downwind from the Pandora site. This makes it possible to improve the statistics and better interpret the high-spatial-resolution measurements made by TROPOMI. By using this wind-based validation technique, the number of coincident measurements can be increased by about a factor of five. Using this larger number of coincident measurements, this work shows that both TROPOMI and Pandora instruments can reveal detailed spatial patterns (i.e., horizontal distributions) of local and transported NO$_2$ emissions, which can be used to evaluate regional air quality changes. The TROPOMI ECCC NO$_2$ research data product





shows improved agreement with Pandora measurements compared to the TROPOMI standard tropospheric $NO_2$ data product, demonstrating benefits from the high-resolution regional air quality forecast model.

## 1 Introduction

Nitrogen dioxide ($NO_2$) is an important air pollutant and plays a critical role in tropospheric photochemistry (e.g., ECCC,
2016; EPA, 2014). It is primarily emitted to the lower troposphere from combustion processes and biomass burning as well as from lightning to the upper troposphere. $NO_2$ forms nitrate aerosol that contributes to acid deposition and eutrophication of lakes (ECCC, 2016). Exposure to $NO_2$ can lead to adverse health effects, such as decrease in lung function and increase in susceptibility to allergens for people with asthma (Anenberg et al., 2018; EEA, 2017; WHO, 2017).

Total vertical column $NO_2$ can be measured by ground-based UV-visible remote sensing instruments using direct-sun, zenith-
sky, or off-axis spectroscopy techniques (Cede et al., 2006; Drosoglou et al., 2017; Herman et al., 2009; Lee et al., 1994; Noxon, 1975; Piters et al., 2012; Roscoe et al., 2010; Vaughan et al., 1997). These measurements are of high quality and good precision, and have been widely used for atmospheric chemistry studies (e.g., Adams et al., 2012; Hendrick et al., 2014) and satellite validations (e.g., Celarier et al., 2008; Drosoglou et al., 2018; Irie et al., 2008; Wenig et al., 2008). Among all these different viewing geometries, direct-sun measurements are of high accuracy, and are not dependent on radiative transfer models
(RTMs) to calculate air mass factors (AMFs) (Herman et al., 2009) or on knowledge of other atmospheric constituents.

The Pandora sun spectrometer is an instrument that measures vertical column densities (total columns) of trace gases in the atmosphere using sun and sky radiation in the UV-visible spectral region. It was developed at the National Aeronautics and Space Administration (NASA) Goddard Space Flight Center and first deployed in the field in 2006 (Herman et al., 2009). One of its primary data products is $NO_2$ total vertical column density ($VCD_{total}$) from the direct-sun viewing mode, where $VCD_{total}$
represents the vertically integrated number of molecules per unit area and is reported in units of molec $cm^{-2}$ or Dobson Unit (1 DU = $2.6870 \times 10^{16}$ molec $cm^{-2}$). The Pandora direct-sun $NO_2$ $VCD_{total}$ products have been validated through many field campaigns (Flynn et al., 2014; Lamsal et al., 2017; Martins et al., 2016; Piters et al., 2012; Reed et al., 2015), ground-based comparisons (Herman et al., 2009; Wang et al., 2010), and satellite validations (Griffin et al., 2019; Ialongo et al., 2016; Lamsal et al., 2014). Since their introduction in 2006, Pandora spectrometers have been deployed at more than 50 sites globally.
Funded by ESA, the Pandonia project (http://pandonia.net) was established in 2015 to provide fiducial reference measurements for satellite instruments. From the collaboration between the NASA Pandora Project (http://pandora.gsfc.nasa.gov) and the ESA Pandonia project, the Pandonia Global Network (PGN) was officially launched in June 2019 (https://www.pandonia-global-network.org/). As a research partner to NASA's Pandora project and ESA's Pandonia project, the Environment and Climate Change Canada (ECCC) Canadian Pandora team carries out Pandora measurements at six Canadian sites (Szykman
et al., 2019). In this work, measurements made at three sites in southern Ontario, Canada are used. These three sites represent different environments in or north of the Greater Toronto Area (GTA).



Using Pandora measurements in and north of the GTA, two versions of TROPOMI tropospheric $NO_2$ data products are evaluated in this work: the standard TROPOMI $NO_2$ (offline v1.1 to v1.2, Boersma et al., 2018; Eskes et al., 2019; Eskes and Eichmann, 2019; van Geffen et al., 2019) processed by the Royal Netherlands Meteorological Institute (KNMI), and the ECCC recalculated TROPOMI $NO_2$ (Griffin et al., 2019). The ECCC recalculated $NO_2$ data (referred to as ECCC $NO_2$) utilize AMFs generated using higher-resolution input for profile shape, albedo, and snow flag. These AMFs were found to have better agreement with aircraft and ground-based measurements in the Athabasca Oil Sands Region (AOSR) (Griffin et al., 2019) than the standard TROPOMI tropospheric $NO_2$ (referred to as KNMI $NO_2$). One part of this work focuses on further comparison between the KNMI and ECCC TROPOMI $NO_2$ data products.

Traditionally, ground-based measurements that are spatially and temporally close are used to validate satellite data (e.g. Boersma et al., 2009; Celarier et al., 2008; Griffin et al., 2019; Herman et al., 2009; Lamsal et al., 2014; Wenig et al., 2008). Depending on the satellite's ground-pixel size (spatial resolution) and orbit, this standard methodology usually has some constraints, such as spatial sampling (satellite data averaging a larger area than the ground-based measurements) and temporal sampling issues (for Sun-synchronous orbits, satellite instruments only measure once per day over most mid-latitude regions). Furthermore, most satellite measurements are sensitive to cloud cover, and thus for a single site, the number of coincident measurements between satellite and ground-based instruments can be very limited. To improve the statistics and interpretation of the high-spatial-resolution measurements made by TROPOMI, a wind-based method was developed, tested, and applied for TROPOMI $NO_2$ validation. The enhanced number of coincident measurements and combined meteorological data provide information about the regional $NO_2$ distribution and transport patterns.

This paper is organized as follows: Section 2 describes the ground-based and satellite measurements of $NO_2$ and the wind field data. In Section 3, the different validation schemes are introduced, with a detailed description of the new wind-based technique. In Section 4, the KNMI and ECCC satellite $NO_2$ data products are evaluated by comparing with ground-based data at three sites. Lastly, in Section 5, several aspects of the wind-based validation work are discussed including sensitivity tests, $NO_2$ spatial distribution and transport patterns, and performance comparison between the Ozone Monitoring Instrument (OMI) and TROPOMI. Conclusions are given in Section 6.

## 2 Datasets

### 2.1 TROPOMI

TROPOMI is the single payload on the Sentinel 5 Precursor (S5P) satellite, which has a Sun-synchronous orbit with overpass time of around 13:30 local solar time (Veefkind et al., 2012). TROPOMI has near full-surface coverage on a daily basis. The instrument contains three spectrometers that cover the ultraviolet-near infrared (UVN) with two spectral bands at 270–500 nm and 675–775 nm, and one spectrometer that covers the shortwave infrared. The UVN detector developed for TROPOMI is a back-illuminated $1024 \times 1024$ pixel frame transfer charge-coupled device (CCD) (Kleipool et al., 2018). The instrument has a high spatial resolution of 7 km $\times$ 3.5 km (along-track $\times$ across-track) at nadir for bands 2-6 (UVN module) (Eskes et al.,



2019). The high spatial resolution of TROPOMI is a major improvement compared to its predecessor, OMI, which has a ground footprint roughly of 13 km × 24 km at nadir (de Graaf et al., 2016).

### 2.1.1 KNMI NO₂

The standard TROPOMI NO₂ retrieval algorithm was developed by KNMI and utilizes the bands of the ultraviolet-near infrared
spectrometer (van Geffen et al., 2019). The retrieval algorithm is based on the NO₂ DOMINO retrieval previously used for OMI spectra (Boersma et al., 2011) with improvements made for retrieval sub-steps (Boersma et al., 2018; van Geffen et al., 2015, 2019; Lorente et al., 2017; Zara et al., 2018). The total NO₂ slant column density (SCD) is retrieved by using the Differential Optical Absorption Spectroscopy (DOAS) method (e.g., Platt, 1994; Platt and Stutz, 2008). The SCD is separated into stratospheric ($SCD_{strat}$) and tropospheric ($SCD_{trop}$) components using information from a data assimilation system (van
Geffen et al., 2019). Next, $SCD_{strat}$ and $SCD_{trop}$ are converted to stratospheric and tropospheric vertical columns, respectively ($VCD_{strat}$ and $VCD_{trop}$) by applying appropriate altitude-dependent AMFs based on a look-up table. The look-up table requires daily information on the vertical profile of NO₂ from the TM5-MP model (at 1° × 1° resolution; Williams et al., 2017) and the surface albedo information derived from a monthly OMI climatology (on a 0.5° × 0.5° resolution; Kleipool et al., 2008). TROPOMI uses a snow flag from the Near real-time Ice and Snow Extent (NISE), and the albedo is set to 0.6 if the surface
beneath is covered in snow or ice. For this study, we use offline (OFFL) level 2 v1.1 to v1.2 (van Geffen et al., 2019), which is the first released offline version of the TROPOMI tropospheric and stratospheric NO₂ columns ([http://www.tropomi.eu](http://www.tropomi.eu)). During preparation of this paper, a new reprocessing level v1.3 product has become available (van Geffen et al., 2019). The total column NO₂ used in this work is the sum of $VCD_{strat}$ and $VCD_{trop}$. Spatial resolution varies with across-track position, and in this study, the average pixel size is about 5.9 km × 7 km. Pixels that are fully or partially covered by clouds were filtered;
here we used 0.3 as a cutoff for the radiative cloud fraction (provided with TROPOMI data).
The TROPOMI NO₂ data product bias and random uncertainty requirements (ESA EOP-GMQ, 2017) are shown in Table 1. Independent preliminary validation by S5P Mission Performance Center (MPC) and S5P validation team concludes that OFFL level 2 NO₂ data is in overall agreement with reference measurements collected from global ground-based networks (Eskes and Eichmann, 2019; Lambert et al., 2019). TROPOMI tropospheric columns were compared with multi-axis DOAS (MAX-
DOAS) data at 14 sites. It was found that TROPOMI tropospheric columns have a median negative bias of less than 50 %. TROPOMI stratospheric columns were compared with zenith-sky scatter-light DOAS (ZSL-DOAS) data and a 0.01 DU negative bias (below 5 %) was found. Total columns were compared with measurements by more than 10 Pandora instruments, and showed a negative bias, with TROPOMI being up to 45 % lower and showing a lower than expected accuracy. However, currently, the random uncertainties of the data product have not been fully accessed.

### 2.1.2 ECCC NO₂

Following Griffin et al., (2019), tropospheric AMFs, which are recalculated at a much higher resolution (10 km × 10 km) than those for the standard TROPOMI product (about 40 km × 110 km in the GTA), are used to produce the ECCC version of



TROPOMI tropospheric $NO_2$ data. The Global Environmental Multiscale - Modelling Air-quality and Chemistry (GEM-MACH) operational model output (version 2, at 10 km × 10 km resolution, the closest hourly data) was used to provide the $NO_2$ profile shape used in the AMF calculation. GEM-MACH is ECCC's regional air quality forecast model. It is run operationally two times per day to predict hourly surface pollutant concentrations over North America for the next 48 hours

(Moran et al., 2009; Pavlovic et al., 2016; Pendlebury et al., 2018). Physical and chemical processes represented in GEM-MACH include emissions, dispersion, gas- and aqueous-phase chemistry, inorganic heterogeneous chemistry, aerosol dynamics, and wet and dry removal. The ECCC AMF calculation used the Interactive Multisensor Snow and Ice Mapping System (IMS) data (Helfrich et al., 2007) to flag pixels with snow cover. Improved albedo inputs were created using averaged monthly MODIS albedo for areas without snow cover and a MODIS climatology for snow-covered areas. With the inputs from

GEM-MACH, MODIS, IMS, and the SASKTRAN radiative transfer model (Bourassa et al., 2008; Dueck et al., 2017; Zawada et al., 2015), new tropospheric AMFs were calculated. More details about this TROPOMI ECCC tropospheric $NO_2$ data product can be found in Griffin et al. (2019). In this work, the ECCC total column $NO_2$ data products are generated by adding ECCC tropospheric columns to KNMI standard stratospheric columns.

## 2.2 OMI

OMI is a Dutch-Finnish nadir-viewing UV-visible spectrometer aboard NASA's Earth Observing System (EOS) Aura satellite that was launched in July 2004. It measures the solar radiation backscattered by the Earth's atmosphere and surface between 270 and 500 nm with a spectral resolution of 0.5 nm (Levelt et al., 2006, 2018). OMI has a 780 × 576 CCD detector that measures at 60 across-track positions simultaneously, and thus, does not require across-track scanning. Due to this approach, the spatial resolution of the CCD pixels varies significantly along the across-track direction: the pixels near the track centre

have a ground footprint of 13 km × 24 km, whereas the pixels close to the track edge (e.g., view zenith angle = 56°) have a ground footprint roughly of 23 km × 126 km (de Graaf et al., 2016). Note that from 2012 onwards, the smallest pixels (across-track positions) can no longer be used and are excluded from the analysis (known as the "row anomaly", i.e. Levelt et al., 2018). This means the "smallest" pixels available for OMI are larger than 13 km × 24 km.

### 2.2.1 SPv3 $NO_2$

The OMI total column $NO_2$ data used in this work are the NASA standard product (SP) (Bucsela et al., 2013; Wenig et al., 2008) version 3.1 level 2 (SPv3.1) (Krotkov et al., 2017). The $NO_2$ SCDs are derived using the DOAS technique in the 405-465 nm window (Marchenko et al., 2015). The AMFs used in SPv3.1 are calculated by using 1° × 1.25° (latitude × longitude) resolution a priori $NO_2$ and temperature profiles from the Global Modeling Initiative (GMI) chemistry-transport model with yearly varying emissions (Krotkov et al., 2017).



### 2.2.2 ECCC NO$_2$

Similar to TROPOMI ECCC NO$_2$, the same alternative tropospheric ECCC AMFs were applied to OMI data. The OMI-ECCC tropospheric column data were evaluated in McLinden et al. (2014), which showed that the OMI-ECCC data has increased the peak NO$_2$ VCD$_{trop}$ occurring over the Canadian AOSR by a factor of two. In this work, the OMI-ECCC total column NO$_2$ data

products are generated by adding ECCC tropospheric columns to OMI standard stratospheric columns. Compared to TROPOMI-ECCC, which uses the hourly GEM-MACH profiles, OMI-ECCC uses the modelled monthly NO$_2$ climatology as the input in the AMFs calculation.

### 2.3 Pandora

The Pandora instrument records spectra between 280 and 530 nm with resolution of 0.6 nm (Herman et al., 2009, 2015;

Tzortziou et al., 2012). It uses a temperature-stabilized Czerny-Turner spectrometer, with a 50 µm entrance slit, 1200 groove mm$^{-1}$ grating, and a 2048 × 64 back-thinned Hamamatsu CCD detector. The spectra are analysed using a total optical absorption spectroscopy (TOAS) technique (Cede, 2019), in which absorption cross sections for multiple atmospheric absorbers, such as ozone, NO2, and sulphur dioxide (SO2), are fitted to the spectra.

The Pandora direct-sun total column NO$_2$ data are produced using Pandora's standard NO$_2$ algorithm implemented in the

BlickP software (Cede, 2019). The measured direct-sun spectra from 400 to 440 nm are used in the TOAS analysis. A synthetic reference spectrum is produced by averaging multiple measured spectra which get corrected for the estimated total optical depth included in it. Cross sections of NO$_2$ at an effective temperature of 254.5 K (Vandaele et al., 1998), ozone at an effective temperature of 225 K (Brion et al., 1993, 1998; Daumont et al., 1992), and a fourth-order polynomial are all fitted. The resulting NO$_2$ SCDs are then converted to total column by using direct-sun geometry AMFs. Herman et al. (2009) showed that Pandora

direct-sun total column NO$_2$ has a clear-sky precision of 0.01 DU (in slant column) and a nominal accuracy of 0.1 DU (in vertical column). Additional information on Pandora calibrations, operation, and retrieval algorithms can be found in Herman et al. (2009) and Cede (2019).

The Pandora instruments nos. 103 and 104  have been deployed in Downsview, Toronto (43.781° N, -79.468° W; suburban) since 2013 to perform direct-sun measurements (Zhao et al., 2016). The instruments are installed on the roof of the ECCC

Downsview building at an altitude of 187 m a.s.l.. The building is located in a suburban area with multiple roads nearby. Since February 2018, the instruments have employed an alternating direct-sun, zenith-sky, and multi-axis observation schedule, which includes direct-sun measurements every 5 minutes during the sunlit period.

The Pandora instruments nos. 108 and 145 have been deployed in Egbert (44.230° N, -79.780° W; rural) and the University of Toronto St. George Campus (43.661° N, -79.399° W, referred to as UTSG; urban), respectively since May 2018. The same

alternating observation schedule is implemented. Pandora no. 108 is located on the roof of the ECCC Center for Atmospheric Research Experiments (CARE) building in Egbert at an altitude of 251 m a.s.l.. The building is in a rural area, which is surrounded by farmlands. Pandora no. 145 is located in the University of Toronto Atmospheric Observatory (TAO) in

downtown Toronto at an altitude of 174 m a.s.l.. A map of the GTA and surrounding areas is shown in Fig. 1, overlaid with a color map of TROPOMI KNMI $NO_2$ tropospheric columns averaged over the March 2018 to March 2019 period utilizing the pixel averaging technique (Fioletov et al., 2011; Sun et al., 2018). It is clear that the three sites (Downsview, Egbert, and UTSG) represent three different $NO_2$ pollution levels.

## 2.4 Wind data

### 2.4.1 ERA-Interim for OMI

As in several previous studies (Fioletov et al., 2017, 2015; McLinden et al., 2016), wind speed and direction data for each satellite pixel from the European Centre for Medium-Range Weather Forecasts (ECMWF) reanalysis data (Dee et al., 2011; http://apps.ecmwf.int/datasets/), i.e., ERA-Interim, were merged with OMI measurements. Wind profiles are available every 6 hours on a 0.75° horizontal grid and are interpolated in time and space to the location of each OMI pixel centre. U and V (west-east and south-north, respectively) wind-speed components were interpolated spatially and temporally to the location and overpass time of each OMI pixel. The wind components were then averaged in the vertical between 1000 and 900 hPa where the majority of the tropospheric $NO_2$ mass resides.

### 2.4.2 ERA-5 for TROPOMI

ERA-5 data has better spatial and temporal resolution (1 hour on a 0.28° horizontal grid, approximately 30 km) than ERA-Interim. Thus, ERA-5 data were selected and merged with TROPOMI $NO_2$ data. Wind profiles were interpolated spatially and temporally for TROPOMI pixels, and 1000-900 hPa vertical pressure levels were used in averaging the wind speed and direction. The results of "ERA-Interim + OMI" and "ERA-5 + TROPOMI" are compared and presented in Section 5.2. The other combination such as "ERA-Interim + TROPOMI" were also evaluated, but it was found that "ERA-Interim + TROPOMI" result did not perform as well as the combination of "ERA-5 + TROPOMI". This is unsurprising since the core of the wind-based method (see Section 3.2) is the quality of high-resolution wind and satellite data. Thus, the "ERA-Interim + TROPOMI" combination was not included in this work.

## 3 Validation schemes

### 3.1 Standard approach

To validate the satellite measurements, coincident ground-based data are required. The coincidence criteria are normally composed of spatial, temporal and quality control criteria (e.g., Boersma et al., 2018; Drosoglou et al., 2017; Griffin et al., 2019; Irie et al., 2008; Toohey and Strong, 2007). For example, in Zhao et al. (2019), the coincidence criteria used to pair ground-based observations (Pandora) and OMI data include: (1) nearest (in time) measurement that was within ± 30 min of OMI overpass time, (2) closest OMI ground pixel (having a distance of less than 20 km from the ground pixel centre to the





location of the Pandora instrument, and (3) cloud fraction <= 0.3 (the effective geometric cloud fraction as determined by the OMCLDO2 algorithm), and only high-quality OMI data are used (VcdQualityFlags = 0) (Celarier et al., 2016). This simple coincident measurements selection scheme is referred here as the "standard" method.

In this work, similar criteria are used with some adjustments. The temporal criterion is changed from ± 30 min to ± 10 min of

TROPOMI overpass time (this is to ensure the standard method can be fairly compared with the new wind-based method, see Section 3.2). Since TROPOMI has better spatial resolution than that of OMI, the selected spatial criterion is set to 10 km for TROPOMI. Similar to OMI, only high-quality TROPOMI data are used (qa_value > 0.75) (Eskes et al., 2019). Pandora direct-sun $NO_2$ total column data of assured high-quality are used in the validation (Cede, 2019).

## 3.2 Wind-based method

To make more use of the high-resolution measurements made by TROPOMI and to improve their validation, a wind-based method is tested, which can increase the number of coincident measurements. In addition to coincident and co-located data, this method looks at upwind (downwind) TROPOMI pixels that will arrive at (have passed over) the Pandora site within a short time window. Technically, this is done using wind rotation and aligning all wind directions to the preferred direction. After the rotation, all ground-pixels have a common effective wind direction, and can be analyzed together regardless of the

true wind direction. Any $NO_2$ source located between the satellite pixel and the Pandora site will affect the performance, and we need to look at the TROPOMI-Pandora differences as a function of the wind direction.

In general, the wind-based method adapts a pixel-rotation technique developed and used in several previous studies (e.g., Fioletov et al., 2015; Pommier et al., 2013). In previous, the pixel-rotation involves a rotation of each satellite ground-pixel around the $SO_2$ source location (e.g., smelters or mining areas). In this work, we adapted the pixel-rotation technique, where

the satellite observations are rotated around a point, which in this case, is the location of the ground-based instrument.

First, the initial coordinates of each satellite pixel (geographic coordinate; $G_{initial}(lat,lon)$) are interpolated to the horizontal distance from the selected centre (local tangent plane coordinate; $P_{inital}(x,y)$, where $x$ is east-west distance and $y$ is north-south distance). Figure 2a shows the initial positions of pixels in the local tangent plane coordinate, where the ground-based instrument site is at $P(0,0)$. Then a rotation matrix $\mathbf{R}$ is applied to satellite ground-pixels, with

the rotation angle equal to the negative of wind direction ($\theta$):

$$P_{rotate} = \mathbf{R}(-\theta)P_{inital}, \tag{1}$$

$$\mathbf{R}(-\theta) = \begin{bmatrix} \cos(-\theta) & \sin(-\theta) \\ -\sin(-\theta) & \cos(-\theta) \end{bmatrix}. \tag{2}$$

After the rotation, each satellite ground-pixel maintains its upwind-downwind character. In other words, after rotation, the new ground-pixel, $P_{rotate}(x,y)$, can be analyzed assuming that the wind always has a constant direction (from "north"

to "south") as shown in Fig. 2. All pixels in Fig. 2 share the same wind direction, but only three colour-coded pixels are selected to show with the wind arrow. Thus, for a given rotated pixel, the closest distance it can reach to the site, $P(0,0)$, is $x_{rotate}$, at a time given by:





$$t_{coincident} = t_{meas} + \frac{y_{rotate}}{v}, \quad\quad\quad (3)$$

where, $t_{meas}$ is the measurement time of the pixel, $y_{rotate}$ is the $y$ value of $P_{rotate}$, and $v$ is the wind speed. Next, the boundaries of coincident measurement selection are defined as

$$|x_{rotated}| \le \rho, \quad\quad\quad (4)$$

where $\rho$ is an arbitrary distance, referred to as rotational-coincident distance. For example, for TROPOMI, we find the optimized $\rho$ value equal to 5 km. Use of a larger $\rho$ value will increase the number of coincident measurements, while the representativeness of coincident measurements (i.e., whether or not the selected satellite pixel can represent the ground-based measurement at a given time) will decrease. Based on sensitivity tests using various $\rho$ values, a balance between number of coincident measurements and representativeness can be achieved. For other satellites with coarse

spatial resolution, the rotational-coincident distance value has to be increased (e.g., approximately equal to the satellite's ground-pixel size).

This method is valid if the trace gas concentrations do not change rapidly over the pixel travel time, $t_{travel}$. The assumptions made here are 1) the local emission patterns (strength and spatial distribution), 2) chemical reactions, and 3) vertical atmospheric dynamics (i.e., boundary layer variation) do not change rapidly during this time period. All these assumptions are

likely to be close to reality for tropospheric $NO_2$ in most areas around local noon. Even for urban areas, the local emissions are relatively stable around local noon (when Sun-synchronous orbiting satellites such as OMI and TROPOMI pass over a given site) compared to morning or evening rush hours. In addition, the $NO_2$ photolysis rate (Dickerson et al., 1982) and boundary layer height (Garratt, 1994) around local noon are also relatively stable compared to morning or evening conditions. For example, using the rotated plane coordinates (Fig. 2b), any pixel within the rotational-coincident boundaries $\pm\rho$

(indicated by the dashed lines), will "overpass" the site at $t_{coincident}$, having an "overpass" distance from the ground pixel centre to the location of the ground-based instrument less than or equal to 5 km. In the application, we give a cutoff value to the pixel travel time, $t_{travel}$ as

$$t_{travel} = \left| \frac{y_{rotate}}{v} \right| \le 1\ hr , \quad\quad\quad (5)$$

which ensures that the assumptions we made (i.e., emission, chemical, and dynamic changes are not significant in this

period) are valid. The colour-coded pixels in Fig. 2 are examples of upwind pixel (green), downwind pixel (yellow), and out-of-field pixel (red). In general, the wind-based method can identify any pixel that has its simulated "trajectory" passing over the ground-based site. However, if there is a major emission source between the TROPOMI pixel and the ground-based instrument, the difference between satellite and ground-based measurements will increase. This feature is observed and can be used to distinguish local and transported pollution (discussed in Section 5.1).





## 4 Validation results

Times series of Pandora, TROPOMI and OMI total column $NO_2$ are shown in Fig. 3. For the Downsview and UTSG sites, local morning and evening rush hour $NO_2$ pollution can be more than 1.5 DU. When compared to downtown UTSG, suburban Downsview is more polluted, which is mainly due to the heavy traffic in the Downsview area (close to several major highways and the major city airport). In contrast, the rural site Egbert shows no sign of increased $NO_2$ during rush hour.

### 4.1 Comparison between standard and new methods

Figure 4 shows the comparison of results obtained using the standard and the new wind-based methods for defining coincident measurements. The standard and wind-based methods show similar performance in terms of correlation coefficient. Although the correlation coefficients (R) decreased slightly for two of the three sites (for the Downsview site, it decreased from 0.75 to 0.71; for UTSG site, it decreased from 0.71 and 0.65), the R increased for Egbert site (from 0.78 to 0.89). Egbert, as a rural area, has much lower $NO_2$ total columns than the values in urban and suburban areas. Compared to Downsview and UTSG, the correlation coefficients between TROPOMI and Pandora data are improved by wind-based method due to increased observations of transported pollution events. In general, the number of coincident measurements (N) increased for all sites by about a factor of 5 (e.g., from 174 to 939 for Downsview) when using the wind-based method.

For Downsview, Figure 4a and 4b show that the multiplicative biases between TROPOMI and Pandora total column $NO_2$ data (indicated by the slopes of the fitted lines, with fixed zero intercept), are -30% and -23% for the standard and wind-based method, respectively. The results for UTSG and Egbert are shown in Fig. 4c to 4f. Similar to Downsview, TROPOMI data show -28% (standard) and -24% (wind-based) multiplicative biases relative to the Pandora at UTSG. However, in contrast to Downsview, where TROPOMI data show negative bias relative to Pandora data, TROPOMI $NO_2$ observations have 10% (standard) and 4 % (wind-based) positive multiplicative biases at Egbert. Other typical satellite validation metrics, including absolute differences, relative mean differences, and regression slopes are provided in Appendix A.

In general, Fig. 4 shows that the larger number of coincident measurements paired by wind-based method maintained similar good quality as the ones paired by the standard method. Further sensitivity tests on the parameters used in the wind-based method are shown in Appendix B. It is expected that the local emissions are more significant than transported $NO_2$ in downtown Toronto, whereas the transported $NO_2$ is more significant than local emissions in Egbert. Thus, the wind-based method's sensitivity is dependent on local pollution patterns. Details about the sensitivity distance are discussed in Appendix B. In general, for individual sites, a unique sensitivity distance should be evaluated and applied to achieve the best balance between the number of coincident measurements and representativeness of the enlarged dataset (i.e., whether or not the expanded dataset can represent the real local or regional conditions).





## 4.2 ECCC products

The ECCC TROPOMI NO$_2$ data product was also compared to the Pandora measurements. The results from the three sites are shown in Fig. 5. A clear difference between the KNMI version and the ECCC version of TROPOMI data is the multiplicative bias. In general, ECCC data, which is based on a model with much higher spatial resolution, show a positive shift of the fitted

slopes for all three sites by roughly 5 to 15 %. For Downsview, ECCC data decreased the multiplicative bias between satellite and Pandora data from 24-28 % to 15-24 %. For UTSG, similar improvement was found as the multiplicative bias decreased from 24-28 % to 13-20 %. However, for the clean background (rural) site in Egbert, ECCC data increased the multiplicative bias from 4-10 % to 14-15 %. Thus, the results show that ECCC data have a positive shift in the total column values compared to KNMI data when compared to Pandora measurements. The ECCC NO$_2$ product has a better representation of the albedo for

snow-covered areas (Griffin et al., 2019). However, for this period of measurements, the snow-coved satellite ground-pixels are too sparse. Future studies will be performed to evaluate the performance of the ECCC data in snow-covered conditions, after accumulating a sufficient number of snow-covered pixels. More comparison results, such as the absolute difference and relative difference between TROPOMI ECCC data and Pandora data are shown in Appendix A. In general, TROPOMI ECCC data have smaller absolute and relative differences (compared with Pandora data) at Downsview and Egbert, but slightly larger

differences at UTSG.

## 5 Discussion

### 5.1 NO$_2$ spatial distribution and transport patterns

One motivation to increase the number of coincident measurements is to study NO$_2$ spatial distribution and local air quality conditions. In Fig. 6, the coincident TROPOMI and Pandora data are grouped by wind direction, and the mean values of each

group are shown as a function of wind direction. For example, Fig. 6a shows the NO$_2$ results from the Downsview site; the purple line with error bars is Pandora no. 104 total columns, the blue line is TROPOMI KNMI total columns, and the red and yellow lines are the TROPOMI tropospheric and stratospheric components. Although there is a clear offset between the purple and blue lines, indicating an offset between TROPOMI and Pandora NO$_2$ total columns, the general pattern between two datasets is similar. Figure 6a reveals several peaks in the mean NO$_2$ total columns at Downsview, such as at wind directions

180° and 240°, which correspond to the directions from downtown Toronto and the major city airport (Toronto Pearson International Airport, referred to here as YYZ by its airport code, the largest and busiest airport in Canada; in the City of Mississauga, 0.72 million population), respectively. In addition, high Pandora NO$_2$ values for the wind direction of 240° may be related to vehicle emissions from a busy local street located about 100 meters from the site. Meanwhile, low mean NO$_2$ values are found in the 270-360° range, which corresponds to the direction of suburban Downsview, a relatively clean area (in

the northern part of the City of Toronto). Here we define the clean wind direction by using TROPOMI stratospheric and tropospheric NO$_2$. For a given site, any direction that has TROPOMI VCD$_{trop}$ ≤ VCD$_{strat}$ is considered a clean wind direction.



In general, for clean wind directions, the mean difference between TROPOMI and Pandora total columns is within ± 0.05 DU and the mean relative differences are typically within ± 20 %. Here, the mean relative difference is defined as

$$\Delta_{rel} = 100 \times \frac{1}{N} \sum_{i=1}^{N} \frac{(M_{1i} - M_{2i})}{(M_{1i} + M_{2i})/2}, \qquad (6)$$

where N is the number of measurements. We select $M_1$ to be TROPOMI measurements, and $M_2$ to be Pandora measurements.

Figure 6b shows the results from TROPOMI ECCC data. It is clear that the offset between TROPOMI ECCC data and Pandora data has decreased for most polluted wind directions.

The results for UTSG and Egbert sites are shown in Figs. 6c to 6f. For Egbert, almost all wind directions are considered clean air directions, except for 180º. These results highlight that compared to the GTA, other nearby small cities, such as Barrie (see Fig. 1, 0.14 million population; 15 km away from Egbert, within the 30° wind direction bin) are not significant $NO_2$ sources

to Egbert. The UTSG site experiences relatively clean air from 60° to 120°. The major $NO_2$ peak at 150° is linked to the direction from the city's central business district (2 km away from measurement site). The second peak at 240° to 270° is likely linked to the direction of a large diesel train yard for the local train service and the YYZ airport (18 km away from measurement site).

The similarity of the $NO_2$ horizontal distribution patterns observed by TROPOMI and Pandora is also evaluated. The

correlation coefficients between TROPOMI (blue lines) and Pandora (purple lines) angular total column $NO_2$ data (as a function of wind direction) are shown in Fig. 6, referred to here as $R_{angle}$. In general, the patterns show high similarity between satellite and ground-based results, with $R_{angle}$ larger than 0.8 for all three sites, and TROPOMI ECCC data have equal or higher correlation coefficients with Pandora data compared to TROPOMI KNMI data.

To further evaluate the agreement between sets of coincident measurements, the mean difference and mean relative difference

between satellite and ground-based results are shown in Fig. 7. The mean differences between TROPOMI and Pandora are within ± 0.1 DU, except for Downsview data from the 240° wind direction. For Downsview, the highest relative difference is found to be -36 % for the 240° wind direction. Similarly, the largest discrepancies between Pandora and TROPOMI at Egbert and UTSG are found at the wind directions with highest $NO_2$ values, such as 240° for UTSG and 180° for Egbert. For the clean air direction, such as 270-360° for Downsview, the mean relative differences are typically within ± 30 %. TROPOMI ECCC

data performed better for the Downsview and UTSG sites, whereas TROPOMI KNMI data performed slightly better for the Egbert site.

The discrepancies between TROPOMI and Pandora mean differences also indicate the types of $NO_2$ sources. A $NO_2$ peak value is more likely from a regionally transported $NO_2$ source (e.g., a few ground-pixels away), if the mean difference between Pandora and TROPOMI is small (i.e., both Pandora and TROPOMI measured the peak). If the mean difference is large (i.e.,

Pandora measured the peak, whereas TROPOMI did not), then the measured $NO_2$ peak is likely from a localized source (e.g., within or around one ground-pixel). For example, in Fig. 7a for Downsview, the 180° peak shows -0.06 DU mean difference, whereas the 240° peak shows -0.13 DU mean difference. Thus, the 240° peak is more influenced by some near-local $NO_2$ source. Similarly, in Fig. 6b at Egbert, the 180° peak shows only -0.005 DU mean difference. Thus this peak is more weighted

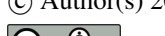



by a far-transported NO$_2$ source. In general, this NO$_2$ distribution study shows that combining Pandora and satellite measurements can be a powerful tool to study local or regional air quality.

The number of coincident pairs and the number of unique days for each wind-bin are shown in Fig. 8. In general, due to the uneven distribution of wind direction, some binned wind directions have a limited number of coincident pairs between

TROPOMI and Pandora, e.g., 60°. Thus, the interpretation of results from these bins is difficult. However, for other bins, such as 180° bin for Egbert, there are 46 coincident measurements from seven days. Thus, we can be confident about the sharp peak signal observed in Figs. 6e and 6f and conclude that for Egbert, the main pollution events are transported from the Toronto area.

## 5.2 Application on medium-resolution satellite (OMI)

The standard and wind-based methods for determining coincidences were also applied to OMI NO$_2$ observations. In this study, we used OMI and Pandora no. 104 measurements from 2015 to 2018 at Downsview. By extending the observation period by a factor of four (i.e., only one year of TROPOMI observations were used in Sections 4 and 5.1, while four years of OMI observations were used here), OMI measurements can reveal similar NO$_2$ spatial distributions to those of TROPOMI (i.e., results in Sect. 5.1).

Figures 9a to 9d show the results of applying the standard and wind-based methods to OMI data. Figures 9a and 9b show the results using the standard method, and the OMI SPv3 NO$_2$ and the OMI ECCC NO$_2$ data products (see Sect. 2.2), respectively. The general performance of OMI SPv3 and OMI ECCC data are similar, and OMI ECCC data have a slightly lower multiplicative bias (28 %) than OMI SPv3 (34 %). Due to the lower spatial resolution of OMI, we modified the coincident criteria used above for TROPOMI. For the wind-based method, the $\rho$ value criteria was changed from 5 km to 20 km, and

$t_{travel}$ was changed from 1 hr to 3 hr, compared to the criteria used for TROPOMI (see Sect. 3.3). The correlations of OMI and Pandora total columns are smaller than those found in Sect. 4 (using TROPOMI and ERA-5 wind field). To make the comparison between OMI and TROPOMI consistent, these modified wind-based criteria ($\rho$ and $t_{travel}$ value) are also applied to TROPOMI data (see Fig. 9e). Similar to other studies (e.g., Eskes and Eichmann, 2019), OMI data show a larger multiplicative bias relative to Pandora than TROPOMI. Although TROPOMI data used in this work only cover one

year and OMI data cover four years, TROPOMI data have about five times the number of coincident measurements compared to OMI data (see Figs. 9c and 9e). In general, the proposed wind-based method is more powerful for high-spatial-resolution satellite instruments than medium-resolution instruments. The same tests were performed on the OMI ECCC data products, as shown in Figs. 9b, 9d, and 9f. Similar to the results in Sect. 4, OMI ECCC and TROPOMI ECCC data have lower multiplicative biases relative to Pandora NO$_2$ total columns than do OMI SPv3 or TROPOMI KNMI data.

Another motivation for applying the wind-based method to OMI is to assess if the spatial distribution of NO$_2$ in this area (Downsview) has had any significant changes over this four-year period. Binning the data by wind direction (see Fig. 10) shows that the NO$_2$ spatial distribution patterns revealed by OMI and TROPOMI are similar; i.e., the main pollution



sources are from the south (from downtown Toronto) and southwest (from the YYZ airport), and clean air from the north (from the suburban area).

For 2018, depending on the wind direction, absolute differences of up to 0.13-0.15 DU can be observed by Pandora and TROPOMI (indicated by Figs. 10b and c). From 2015 to 2018, absolute differences of up to 0.17 DU can be seen for Pandora, and 0.12 and 0.13 DU for OMI SPv3.1 and OMI ECCC, respectively (indicated by Figs. 10c and d). In general, OMI data for 2015-2018 and TROPOMI data for 2018 demonstrate a similar dependence on the wind direction.

**5.3 Precision and accuracy**

To further assess the quality of TROPOMI KNMI and TROPOMI ECCC $NO_2$ data products, and to determine if they meet the TROMPOMI design bias and precision requirements (Eskes and Eichmann, 2019), we performed statistical uncertainty and bias estimations for TROPOMI and Pandora data. In general, by comparing the same quantity retrieved from different remote sensing instruments, we can characterize the differences between them, which are a combination of random uncertainties and systematic bias. Theoretically, information about the random uncertainties can be derived from the measurements (Fioletov et al., 2006; Grubbs, 1948; Toohey and Strong, 2007; Zara et al., 2018; Zhao et al., 2016, 2019a).

As an example, we define the two measured $NO_2$ total column data (denoted as $M_1$ and $M_2$, for Pandora nos. 103 and 104 measurements, respectively) as simple linear functions of the true $NO_2$ total column value ($X$) and instrument random uncertainties ($\delta_1$ and $\delta_2$), and assume that there is no multiplicative or additive bias between two Pandora datasets, giving

$$M_1 = X + \delta_1$$
$$M_2 = X + \delta_2 \,. \qquad (7)$$

Note these two Pandora instruments are located at the same site, i.e., Downsview. If we assume that the instrument random uncertainties are independent of the measured $NO_2$ total column, the variance of $M$ is the sum of the variances of $X$ (around the mean of the dataset) and $\delta$,

$$\sigma_{M_1}^2 = \sigma_X^2 + \sigma_{\delta_1}^2$$
$$\sigma_{M_2}^2 = \sigma_X^2 + \sigma_{\delta_2}^2 \,. \qquad (8)$$

If the difference between two Pandoras does not depend on $X$ (no multiplicative bias), and the random uncertainties of the two instruments are not correlated, then the variance of the difference is equal to the sum of the variance of the random uncertainties,

$$\sigma_{M_1 - M_2}^2 = \sigma_{\delta_1}^2 + \sigma_{\delta_2}^2 \,. \qquad (9)$$

Then, the variance of the instrument random uncertainties can be solved by

$$\sigma_{\delta_1}^2 = \left( \sigma_{M_1}^2 - \sigma_{M_2}^2 + \sigma_{M_1 - M_2}^2 \right)/2$$
$$\sigma_{\delta_2}^2 = \left( \sigma_{M_2}^2 - \sigma_{M_1}^2 + \sigma_{M_1 - M_2}^2 \right)/2 \,. \qquad (10)$$

Equation (10) can be used to estimate the standard deviation of instrument random uncertainties ($\sigma_{\delta_1}$ and $\sigma_{\delta_2}$). The variances $\sigma_{M_i}^2$ and $\sigma_{M_1 - M_2}^2$ can be estimated from the available measurements (with some uncertainty). The uncertainties of the $\sigma_{\delta_1}^2$ and



$\sigma_{\delta_2}^2$ estimates depend on the sum of all three variances $\sigma_{M_1}^2$, $\sigma_{M_2}^2$, and $\sigma_{M_1-M_2}^2$, and can be high even if the estimated variance itself is low (but one or more of the variances $\sigma_{M_1}^2$, $\sigma_{M_2}^2$, and $\sigma_{M_1-M_2}^2$ are high). Thus, the estimates are only as accurate as the least accurate of these parameters. Following the method in Zhao et al. (2016), the variance estimates can be improved by increasing the number of data points or by reducing variance of $X$ by removing some of its natural variability. Thus, the $M_1$

and $M_2$ used in the statistical uncertainty estimation are replaced by so-called residual $NO_2$, which is defined as the difference between the measured $NO_2$ total column and its daily mean. Figure 11 shows the scatter plots for residual $NO_2$ total columns from Pandora nos. 103 and 104. The model estimated $NO_2$ total column random uncertainties ($U_{Pandora103}$ and $U_{Pandora104}$) for two instruments are the same, 0.02 DU, indicating the good consistency between the two co-located instruments. Compared to the TROPOMI $NO_2$ total column random uncertainty requirement (0.032 DU, see Table 1), this result shows Pandora

instruments have sufficient precision for the TROPOMI $NO_2$ data product validation work.

The statistical uncertainty estimation model was also applied to TROPOMI $NO_2$ total column data. Note that the dataset used is the TROPOMI (both KNMI and ECCC products) and Pandora coincident $NO_2$ total column data, paired by the wind-based method. Details of TROPOMI statistical uncertainty calculation are shown in Appendix B. The results are summarized in Fig. 12, which indicate that Pandora $NO_2$ data have lower random uncertainties than TROPOMI $NO_2$ data for all sites. For example,

the first column in Fig. 12a shows the Pandora $NO_2$ measured at Downsview has 0.03 DU random uncertainty (red cross sign with error bar), which is better than the Pandora total column $NO_2$ nominal accuracy (0.05 DU at 1-sigma level, e.g. (Zhao et al., 2019b)). At Downsview, the TROPOMI KNMI and TROPOMI ECCC total column data products have random uncertainties of 0.05 DU (red square with error bar, Fig. 12a) and 0.06 DU (blue square with error bar, Fig. 12b), respectively. The mean of the reported TROPOMI KNMI total column precision is 0.06 DU at this site (the black square with error bar, Fig.

12a). The black dashed line shows the TROPOMI total column data product precision requirement. The green dashed line shows the Pandora precision that was estimated using two co-located Pandoras at Downsview (Pandora nos. 103 and 104). Note that the estimates, which use the statistical random uncertainty estimation method, are only as accurate as the least accurate of these two instruments. Thus, the statistical model indicates that Pandora has a 0.03 DU precision when compared with TROPOMI, while Pandora has a 0.02 DU precision when compared with another co-located Pandora. The KNMI reported

precisions show the satellite data product has better precision at Egbert (0.03 DU) than at Downsview and UTSG. The statistical uncertainty estimation also shows similar results for TROPOMI $NO_2$ total column data (i.e., $U_{Downsview} > U_{UTSG} > U_{Egbert}$), but about 0.01 DU lower than reported precisions. The TROPOMI slant column has a reported random uncertainty on the order of about 0.022 DU (Eskes and Eichmann, 2019). The reported random uncertainty for the tropospheric column is then derived by dividing the slant column by the tropospheric AMF. Because the tropospheric AMF at Egbert is larger, the derived vertical

column random uncertainty will be smaller. Thus, the changes between the three sites are (at least partly) reflecting differences in tropospheric AMF at these sites.

In general, this result indicates the good quality of TROPOMI reported precision. The TROPOMI $NO_2$ total column data products, however, did not meet the design random uncertainty requirement (0.032 DU, see Table 1), except for the clean site (Egbert). On the other hand, although the TROPOMI KNMI data products have higher multiplicative biases than TROPOMI





ECCC data products, their random uncertainties are lower by 0.01 DU at Downsview and UTSG, and by 0.003 DU at Egbert (Fig. 12, red squares compared with blue squares). However, this result should be taken with caution since TROPOMI and Pandora do not directly measure the same quantities. Pandora measures $NO_2$ slant columns along the line-of-sight between the instrument and the sun, while TROPOMI measures slant columns from a mixture of scattering optical paths. Then, both are

converted into vertical columns. Thus, the statistical uncertainty model estimated random uncertainties (red and blue symbols in Fig. 12) are an upper limits of the TROPOMI total column random uncertainty, since the mismatch of the air masses observed between TROPOMI and Pandora (representativity error) will also produce a random-like signal which adds to the estimate. Moreover, the lower spatial resolution of the parameters used in the KNMI AMF calculations may lead to more uniform retrieved values, i.e. to a lower variability of the retrieved $NO_2$ values, and therefore, lower estimated uncertainties.

Besides precision, the bias of the data is estimated for total column and tropospheric column data products. In Section 4, Figs. 4 and 5 show the TROPOMI KNMI and ECCC total column data have negative multiplicative biases up to 30% and 25% (negative relative difference up to 26% and 19%, see Appendix A), respectively. These results are slightly better than the finding from the S5P NIDFORVAL (NItrogen Dioxide and FORmaldehyde Validation) project, in which the $NO_2$ total column comparisons with more than 10 Pandora instruments showed TROPOMI has a negative bias (up to 45% lower) and a lower

than expected accuracy (Eskes and Eichmann, 2019). Note that the bias is strongly site dependent and will depend on the local gradients in $NO_2$ around the measurement site and the ability of the coarse global TM5-MP model (used by TROPOMI KNMI data, 1° resolution) to produce realistic profiles for individual sites. Apparently the AMF produced by the TM5-MP model has good performance in Egbert, away from the city, but has negative biases inside the City of Toronto. For the tropospheric column data, both TROPOMI KNMI and TROPOMI ECCC products meet the design bias requirement; KNMI and ECCC

tropospheric $NO_2$ columns have a negative multiplicative bias relative to the Pandora measurements of up to 41% and 34%, respectively (see Appendix B).

## 6 Conclusion

This work assessed the quality of the TROPOMI $NO_2$ standard data product developed by KNMI and a TROPOMI $NO_2$ research product developed by ECCC. It was found that both TROPOMI $NO_2$ total column data products met the design bias

requirement. Using the statistical uncertainty estimation, the estimated TROPOMI upper limit precision falls below the design requirement at Egbert, but is above this value at the other two sites. Note that the mismatches due to 1) the difference in the line-of-sight between TROPOMI and Pandora, and 2) the TROPOMI averaged $NO_2$ signals (within 5.9 km × 7 km footprint) over a larger area will both add a random component to the comparisons. The TROPOMI KNMI total column data have 24-28 % negative multiplicative bias at the suburban site (Downsview), and 23-25 % negative bias at the urban site (UTSG).

However, the data show 8-11 % positive bias at the rural site (Egbert). In contrast, the TROPOMI ECCC total column data show improvement with decreased multiplicative biases of 14-20 % and 7-18 % at Downsview and UTSG respectively. However, the bias between Pandora and TROPOMI ECCC data increased to 16-19 % at Egbert. The TROPOMI KNMI and





ECCC total column data have 0.02 to 0.06 DU precision at different sites. In general, benefitting from using the high-resolution (both spatial and vertical) regional air quality forecast model in the AMF calculation, the TROPOMI ECCC research data product shows improved agreement with Pandora instruments compared to the TROPOMI standard tropospheric $NO_2$ data product. It was also found that Pandora data have at least 0.01 to 0.02 DU higher precision than TROPOMI data. Thus, Pandora

instruments are suitable and sufficient for validation of TROPOMI $NO_2$. These findings will help the evaluation and algorithm adjustment work for future TROPOMI $NO_2$ data products. In future, in order to close the uncertainty estimate analysis, a quantification of the variability of $NO_2$ within the TROPOMI footprint would be needed (e.g. aircraft mapping studies can be used to provide such information).

The wind-based validation method used in this work is based on high-spatial-resolution satellite measurements and wind

reanalysis data, and can be applied to future high-spatial-resolution geostationary satellite validation work. This study shows that, by using the wind-based method, the high-resolution satellite instrument not only can reveal fine pollution sources, but also it can reveal the regional and local pollution transport patterns that can be used to identify pollution sources that affect air quality at a particular location. For example, we found that high $NO_2$ events observed at Egbert only occur for a 180° wind direction, corresponding to transported pollutants from Toronto. In addition, no significant local sources were found at Egbert,

and the local background $NO_2$ level from other clean air directions (e.g., north) are below 0.2 DU. In contrast, the wind-direction-based $NO_2$ distributions at Downsview indicate that the enhanced $NO_2$ total columns at this site are linked to both local and transported $NO_2$ pollution. The local background $NO_2$ total columns at Downsview is above 0.3 DU. The downtown Toronto site, UTSG, has more localized $NO_2$ pollution, as expected. However, the $NO_2$ spatial distribution at UTSG shows stronger dependency on the wind direction, and larger gradient than other sites (e.g., from 150° to 210° wind direction, the

mean $NO_2$ decreased from 0.4 to 0.25 DU).

In addition, the wind-based method reveals that the TROPOMI ECCC data show better agreement with Pandora data, especially at wind directions associated with high $NO_2$ levels. This result indicates that the ECCC-recalculated high-spatial-resolution AMFs performed better in capturing the enhanced local $NO_2$ signal. In general, the TROPOMI ECCC product has advantages such as: 1) high-spatial-resolution a priori, 2) high-spatial-resolution albedo map, and 3) improved snow-ice flag.

The standard TROPOMI product has advantages such as radiance closure, which involves the use of the same albedo in the AMF and in the cloud retrieval, such that there is a consistency between AMF radiance levels observed by TROPOMI. At present, the TROPOMI algorithm development team is exploring two aspects to reduce the low bias seen in TROPOMI: 1) for Europe, a similar approach to that used for the TROPOMI ECCC product will be implemented, based on hourly CAMS regional model profiles available at 0.1° resolution (also about 10 km), and 2) cloud pressures: $NO_2$ retrievals based on different

cloud products, e.g. $O_2A$ cloud pressure (Fresco) vs $O_2$-$O_2$ cloud pressure will be evaluated. In future, improvement of Fresco and implement of $O_2$-$O_2$ for TROPOMI will benefit from the correction of bias in TROPOMI $NO_2$ data.

This work also explored the applicability of the wind-based validation method to a medium-resolution satellite instrument (i.e., OMI). Using four years of Pandora and OMI data, we found that the local $NO_2$ distribution and transport patterns have not changed significantly at Downsview. Overall, this work proposed and evaluated new methodologies to assess and validate





satellite observations with ground-based measurements, and provided a detailed assessment of TROPOMI and Pandora $NO_2$ data products.

*Data availability.* Pandora data are available from the Pandonia network (http://pandonia.net/data/). OMI $NO_2$ SPv3.1 data are

available from https://disc.gsfc.nasa.gov/. Any additional data may be obtained from Xiaoyi Zhao (xiaoyi.zhao@canada.ca). TROPOMI data can be downloaded from https://s5phub.copernicus.eu; OMI data are available at https://aura.gesdisc.eosdis.nasa.gov/data/Aura_OMI_Level2/OMNO2.003/. TROPOMI ECCC research product is available at http://collaboration.cmc.ec.gc.ca/cmc/arqi/.

*Author contributions.* XZ analyzed the data and prepared the manuscript, with significant conceptual input from DG, VF, and CM, and critical feedback from all co-authors. JD, AO, VF, XZ, and SCL operated and managed the Canadian Pandora network. CM and DG generated the TROPOMI and OMI ECCC data products. AC, MT, and MM operated the Pandonia network and provided critical technical support to the Canadian Pandora measurement program and subsequent data analysis. FB and HE provided TROPOMI KNMI data products. KB and KS operated and provided technical support to Pandora

measurements at the UTSG site.

*Acknowledgements.* We thank Ihab Abboud and Reno Sit from ECCC, Orfeo Colebatch from the University of Toronto, and Daniel Santana Diaz and Manuel Gebetsberger from Pandonia for their technical support of Pandora measurements. We acknowledge the NASA Earth Science Division for providing OMI $NO_2$ SPv3.1 data. The Sentinel 5 Precursor TROPOMI

Level 2 product is developed with funding from the Netherlands Space Office (NSO) and processed with funding from the European Space Agency (ESA).

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





## Appendices

### A. Validation metrics and results

Additional validation comparisons were performed to evaluate the quality of TROPOMI NO$_2$ total column data. Tables A1 to A4 present the absolute and relative mean biases between TROPOMI and Pandora calculated for each site. Here, the mean absolute difference is given by

$$\Delta_{abs} = \frac{1}{N} \sum_{i=1}^{N} (M_{1i} - M_{2i}), \qquad\qquad (A1)$$

where N is the number of coincident measurements, $M_1$ is TROPOMI NO$_2$ total column, and $M_2$ is Pandora NO$_2$ total column. To ensure that this work can be directly compared with other recent Pandora-based satellite validation studies (e.g., Herman et al., 2019; Ialongo et al., 2019), two different types of mean relative difference and several slopes based on different regression methods are calculated. The regression methods used include simple linear regression (SLR), zero intercept regression (ZIR), reduced major axis regression (RMA), and orthogonal linear regression (OLR).

The type-1 mean relative difference, defined with respect to the average of the coincident measurements, is given by

$$\Delta_{rel-1} = 100\% \times \frac{1}{N} \sum_{i=1}^{N} \frac{(M_{1i} - M_{2i})}{(M_{1i} + M_{2i})/2} . \qquad\qquad (A2)$$

The type-2 mean relative difference, defined with respect to Pandora measurement, is given by

$$\Delta_{rel-2} = 100\% \times \frac{1}{N} \sum_{i=1}^{N} \frac{(M_{1i} - M_{2i})}{M_{2i}} . \qquad\qquad (A3)$$

To provide a general assessment of the data quality, the validation results are summarized in Tables A1 (TROPOMI KNMI vs. Pandora, using the standard approach), A2 (TROPOMI ECCC vs. Pandora, using the standard approach), A3 (TROPOMI KNMI vs. Pandora, using the wind-based method), and A4 (TROPOMI ECCC vs. Pandora, using the wind-based method).

### B. Sensitivity tests

Sensitivity tests were performed to find the optimized values (e.g., $\rho$ and $t_{travel}$ limits, see Equations 9 and 10) that can be used in the wind-based method for determining measurement coincidences. Figure B1 shows an example of sensitivity tests done for measurements at Downsview. In the test, the coincident data (TROPOMI and Pandora) are further collected into five groups based on their distance to the site (i.e., $y_{rotate}$ value), from 0 to 10 km, 10 to 20 km, 20 to 30 km, and etc. For each group, the mean of difference between TROPOMI and Pandora data is shown in Fig. B1a; the correlation coefficient is shown in Fig. B1b; the slope is shown in Fig. B1c; the number for coincident measurements is shown in Fig. B1d. Figure B1 shows that with extended radius, coincident measurements found by using wind-based method decreased in quality (i.e., increased difference and bias, and decreased correlation). Also, in the sensitivity test, we used a 2-hour pixel travel time limit ($t_{travel} <= 2$ hr, see Section 3.3) to filter out the data transported from long distances. In general, including coincident data from a larger radius (e.g., radius larger than 30 km) do not always contribute much useful information for the validation.

The same tests were performed for UTSG (Fig. B2) and Egbert (Fig. B3). Based on these tests, for the wind-based method, we only use satellite ground-pixels within 30 km. Further tests related to the transport time were performed, such as changing the





pixel travel time limit ($t_{travel}$) to 1 hr or 3 hr (not shown here). The tests indicate that setting the pixel travel time limit to 1 hr ($t_{travel}$ <= 1 hr) can provide sufficient coincident data and the general quality of the data is better with a shorter time limit. Thus, the data shown in this work from wind-based method (in Section 4) all use the same criteria: 1 hr time limit and transport distance within 30 km.

One important message from Figs. B1 to B3 is that the sensitivity distance for each site is different. For example, for TROPOMI KNMI data in Fig. B1b show that for Downsview, the correlation coefficients between TROPOMI and Pandora data drop from 0.70 (0-10 km bin) to 0.35 (20-30 km bin), and then became stable for the 30-40 km and 40-50 km radius bins. However, Fig. B3a (Egbert) shows an increase of correlation from 0.61 to 0.80 in the first three radius bins and Figure B2a (UTSG) shows a sharper decrease of correlation from 0.67 (10-20 km) to 0.32 (20-30 km), and then a decrease to a negative correlation in the
very last radius bin (40-50 km). These features indicate different local $NO_2$ emission and transport patterns. For each correlation curve, the shaper decrease of correlation indicates that those coincident measurements found by the wind-based method start to lose its representativeness; in other words, the assumptions we made in Sect. 3.3 start to lose their validity for pixels that are too far away from the site. However, this sensitivity distance varies from site to site, which depends on the relative weights between local emission and transport of $NO_2$. For clean sites, such as Egbert, transported $NO_2$ is the major
source of pollutants. Thus, it shows a longer sensitivity distance. For urban sites, such as UTSG, which sits inside a localized polluted region, the pixels from a far distance (several pixels away) do not represent the local conditions; in other words, the local emission is the dominant source of $NO_2$. More interestingly, the suburban Downsview site has a mixture of sources. The local highways provide strong local emission $NO_2$ signals, whereas the city urban area and airport provide strong transported $NO_2$ signals.

The comparison between KNMI and ECCC TROPOMI data also reveals some insights into the local air quality differences between these three sites. For example, KNMI and ECCC data show an almost consistent bias at Downsview and Egbert (see Figs. B1c and B3c) for every radius bin. However, Fig. B2c shows that the slopes of KNMI and ECCC data merged at 20-30 km radius, at UTSG. This result indicates the high-resolution model used in ECCC data led to very different AMFs at the city centre compared to the surrounding areas. In general, for Toronto city centre and suburban areas, TROPOMI ECCC data show
better agreement with Pandora $NO_2$ total columns. However, its increased bias at the rural site still needs more investigation.

### C.  Precision and accuracy

TROPOMI ECCC data only include tropospheric $NO_2$; the total column was calculated as the sum of ECCC tropospheric $NO_2$ and KNMI stratospheric $NO_2$. The precision of TROPOMI and Pandora total column $NO_2$ data are estimated using the statistical uncertainty estimation model,

$$\delta_{TROPOMI} = \sqrt[2]{\frac{1}{2}\left(\sigma_{TROPOMI}^2 - \sigma_{Pandora}^2 + \sigma_{TROPOMI-Pandora}^2\right)} \qquad (C1)$$





$$\delta_{Pandora} = \sqrt[2]{\frac{1}{2}(\sigma_{Pandora}^2 - \sigma_{TROPOMI}^2 + \sigma_{TROPOMI-Pandora}^2)} \qquad \text{(C2)}$$

where $\sigma_{TROPOMI}^2$ is the variance of TROPOMI residual $NO_2$, and $\sigma_{Pandora}^2$ is the variance of Pandora residual $NO_2$, and the $\sigma_{TROPOMI-Pandora}^2$ is the variance of their difference. The residual $NO_2$ (see Fig. 11) is calculated by using total columns subtract the daily mean value. Use of the residual $NO_2$ instead of column $NO_2$ is to remove the influence of daily variations.

In Fig. 12, TROPOMI KNMI reported precisions of stratospheric and tropospheric $NO_2$ are used to calculate the reported precision of total column (see the black squares in Fig. 12). The ECCC reported precision of total column is calculated as quadrature of ECCC tropospheric $NO_2$ precision and KNMI stratospheric $NO_2$ precision.

To better understand the random uncertainty budget, the tropospheric and stratospheric random uncertainties are shown in Fig. C1. The TROPOMI $NO_2$ data product random uncertainty requirements for stratospheric and tropospheric are 0.019 and 0.026

DU, respectively (Eskes and Eichmann, 2019). The mean of the reported values for tropospheric and stratospheric columns are shown in Fig. C1 as blue and red squares. The details of TROPOMI ECCC tropospheric $NO_2$ random uncertainty calculation can be found in McLinden et al. (2014). The black square in Fig. C1b is the statistical uncertainty model estimated random uncertainty for TROPOMI KNMI stratospheric data. Since we do not have a Pandora stratospheric $NO_2$ data product, this estimation was made by using Pandora measurements in Egbert at clean air conditions (see Sect. 5.1, i.e., excluding

measurements when the wind direction is from 90° to 270°). Comparing the results from Fig. 12 and C1, it is seen that the upper limit of TROPOMI total column $NO_2$ data products did not meet the requirement because of the high random uncertainties in the tropospheric columns.

The bias of TROPOMI tropospheric $NO_2$ column data has been evaluated by comparison with estimated Pandora tropospheric $NO_2$ column data. In this work, Pandora tropospheric $NO_2$ columns are estimated by

$$VCD_{trop.Pandora} = VCD_{total.Pandora} - VCD_{strat.TROPOMI} \qquad \text{(C3)}$$

where $VCD_{strat.TROPOMI}$ is the TROPOMI stratospheric column that is coincident (selected by both the standard and wind-based methods) with the corresponding Pandora total column. Figures C2 and C3 show the scatter plots of TROPOMI (KNMI and ECCC) vs. Pandora tropospheric columns. Using the slope of zero-intercept fitting as a proxy for bias, we found KNMI data has -41 to 10 % multiplicative bias, ECCC data has -34 to 28 % multiplicative bias. This result indicates that both TROPOMI

KNMI and TROPOMI ECCC $VCD_{trop}$ data products meet the design bias requirement (25 to 50 %, for $NO_2$ tropospheric column).



**Table 1. TROPOMI NO₂ data product requirements extracted from the S5P Calibration and Validation Plan (ESA EOP-GMQ, 2017).**

| Data product | Bias | Random |
|---|---|---|
| Stratospheric column $NO_2$ | < 10 % | 0.019 DU |
| Tropospheric column $NO_2$ | 25-50 % | 0.026 DU |
| Total column $NO_2$ | n.a. | 0.032 DU |





**Table A1. TROPOMI KNMI vs. Pandora total column NO₂, using the standard approach.**

| Pandora serial no. (site) | $\Delta_{abs}$ [DU] | $\Delta_{rel-1}$ [%] | $\Delta_{rel-2}$ [%] | N | R | Slopes | | | |
|---|---|---|---|---|---|---|---|---|---|
| | | | | | | SLR[a] | ZIR[b] | RMA[c] | OLR[d] |
| 104 (Downsivew) | -0.08±0.01 | -25.25±1.82 | -20.40±1.46 | 174 | 0.75 | 0.46 | 0.7 | 0.61 | 0.53 |
| 145 (UTSG) | -0.07±0.01 | -17.89±2.89 | -12.39±2.41 | 130 | 0.71 | 0.48 | 0.72 | 0.67 | 0.58 |
| 108 (Egbert) | 0.02±0.01 | 13.48±2.06 | 19.18±4.06 | 116 | 0.78 | 0.86 | 1.1 | 1.1 | 1.14 |

[a]Simple Linear Regression; [b]Zero Intercept Regression; [c]Reduced Major Axis regression; [d]Orthogonal linear regression. The errors shown for $\Delta_{abs}$, $\Delta_{rel-1}$, and $\Delta_{rel-2}$ are the standard error.

5   **Table A2. TROPOMI ECCC vs. Pandora total column NO₂, using the standard approach.**

| Pandora serial no. (site) | $\Delta_{abs}$ [DU] | $\Delta_{rel-1}$ [%] | $\Delta_{rel-2}$ [%] | N | R | Slopes | | | |
|---|---|---|---|---|---|---|---|---|---|
| | | | | | | SLR | ZIR | RMA | OLR |
| 104 (Downsivew) | -0.06±0.01 | -18.51±2.10 | -14.02±1.79 | 174 | 0.73 | 0.49 | 0.76 | 0.68 | 0.59 |
| 145 (UTSG) | -0.02±0.01 | -5.46±3.16 | 0.36±2.94 | 130 | 0.70 | 0.68 | 0.88 | 0.97 | 0.96 |
| 108 (Egbert) | 0.03±0.01 | 14.09±2.07 | 18.97±3.11 | 116 | 0.84 | 1.07 | 1.15 | 1.28 | 1.34 |

**Table A3. TROPOMI KNMI vs. Pandora total column NO₂, using the wind-based method.**

| Pandora serial no. (site) | $\Delta_{abs}$ [DU] | $\Delta_{rel-1}$ [%] | $\Delta_{rel-2}$ [%] | N | R | Slopes | | | |
|---|---|---|---|---|---|---|---|---|---|
| | | | | | | SLR | ZIR | RMA | OLR |
| 104 (Downsivew) | -0.07±0.01 | -25.71±0.89 | -19.94±0.81 | 939 | 0.71 | 0.65 | 0.77 | 0.92 | 0.89 |
| 145 (UTSG) | -0.05±0.01 | -15.92±1.09 | -10.85±1.04 | 774 | 0.65 | 0.44 | 0.77 | 0.67 | 0.56 |
| 108 (Egbert) | 0.02±0.01 | 11.34±0.76 | 14.54±1.09 | 626 | 0.89 | 0.77 | 1.04 | 0.86 | 0.85 |

**Table A4. TROPOMI ECCC vs. Pandora total column NO₂, using the wind-based method.**

| Pandora serial no. (site) | $\Delta_{abs}$ [DU] | $\Delta_{rel-1}$ [%] | $\Delta_{rel-2}$ [%] | N | R | Slopes | | | |
|---|---|---|---|---|---|---|---|---|---|
| | | | | | | SLR | ZIR | RMA | OLR |
| 104 (Downsivew) | -0.05±0.01 | -19.47±1.06 | -13.56±0.97 | 939 | 0.71 | 0.81 | 0.85 | 1.13 | 1.19 |
| 145 (UTSG) | -0.04±0.01 | -12.86±1.31 | -6.37±1.26 | 774 | 0.55 | 0.44 | 0.80 | 0.81 | 0.68 |
| 108 (Egbert) | 0.03±0.01 | 13.54±0.79 | 17.53±1.25 | 626 | 0.92 | 1.05 | 1.13 | 1.14 | 1.16 |





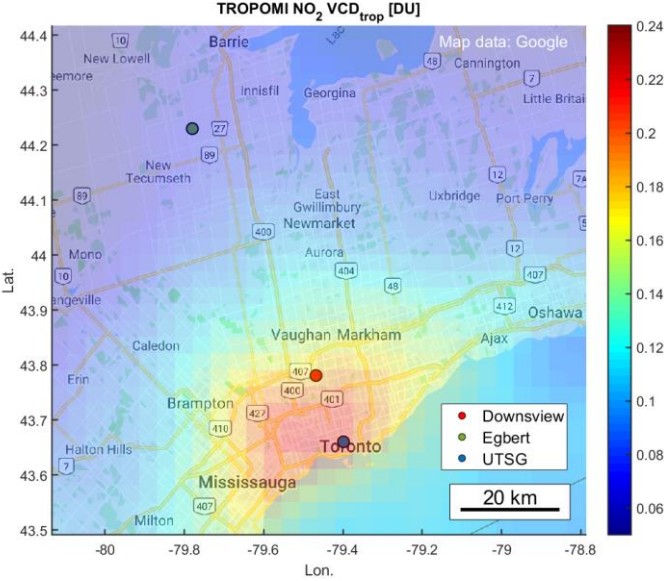

**Figure 1. Pandora sites in and north of the Greater Toronto Area. Colour dots indicate the sites. The map (© Google Maps) is masked with TROPOMI KNMI NO₂ tropospheric columns smoothed by pixel averaging (March 2018 to March 2019).**





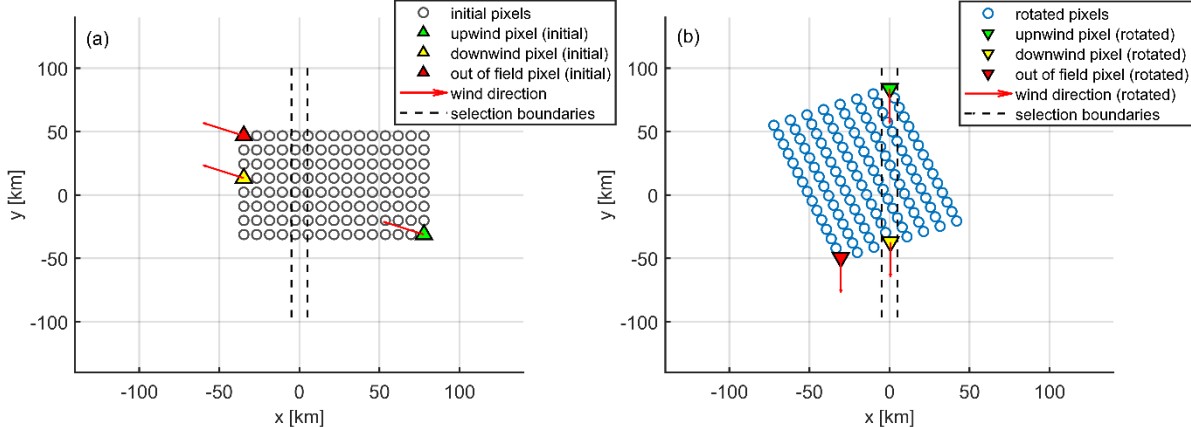

**Figure 2. Visualization of the wind-based method, (a) initial positions of satellite ground pixels, (b) rotated positions of these satellite ground pixels both in local tangent plane coordinates. Examples of upwind (green), downwind (yellow), and out-of-field (red) pixels are shown by colour-coded triangles with wind arrows. Dashed black lines are the boundaries used to select pixels (and measurements) that are coincident with the ground-based site.**





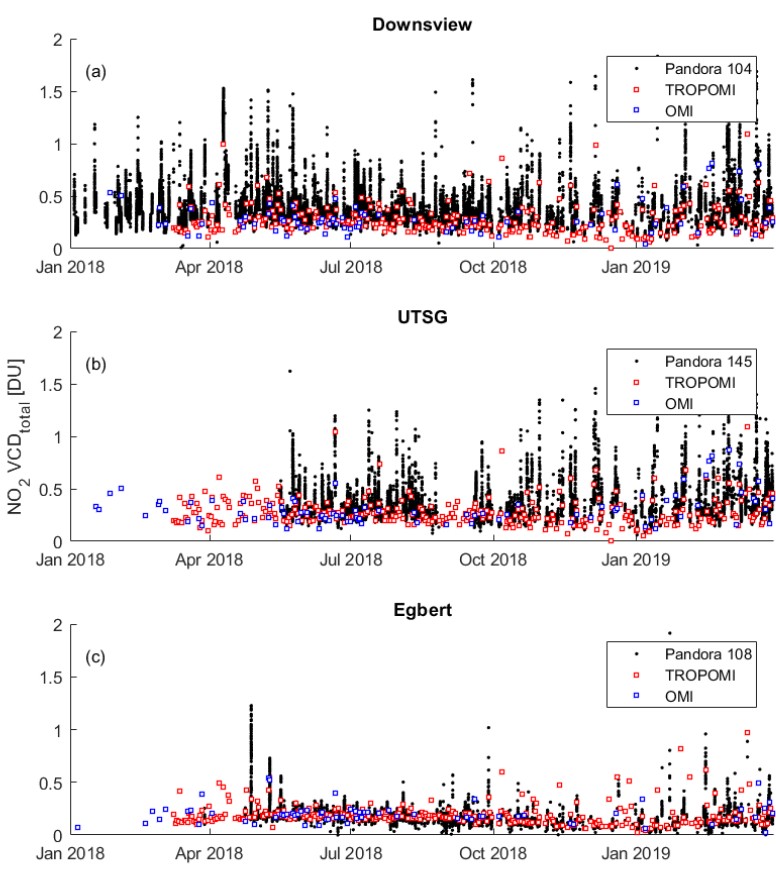

**Figure 3. Time series of Pandora, TROPOMI, and OMI total column NO₂ in the Greater Toronto area.**





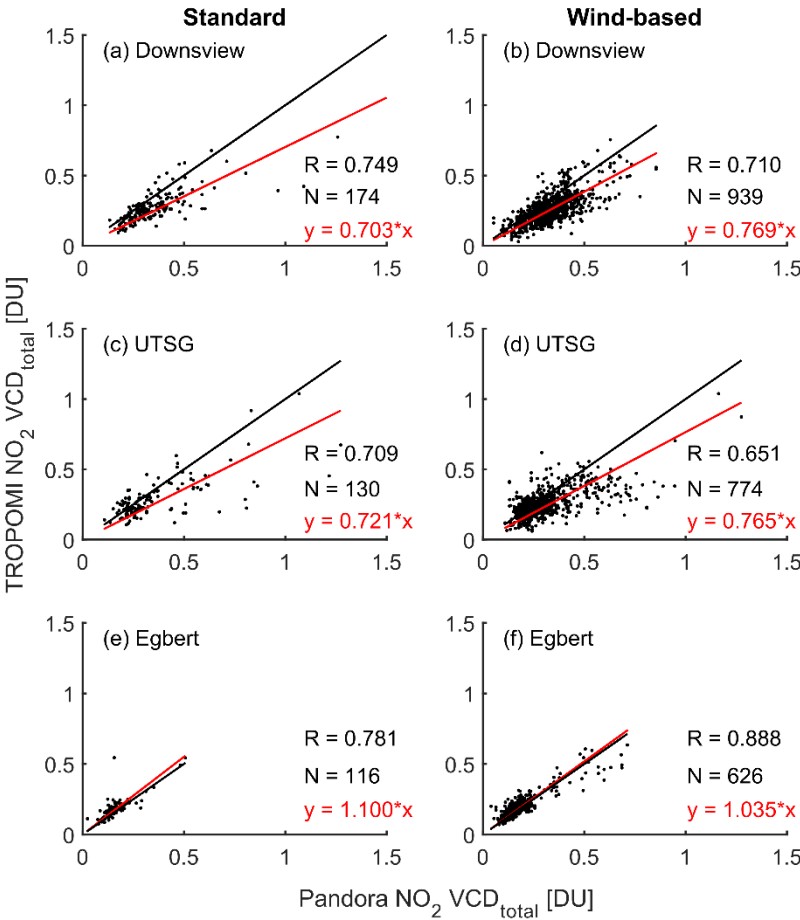

**Figure 4. TROPOMI (KNMI) vs. Pandora NO₂ total columns measured at Downsview, UTSG, and Egbert, using the standard (a, c, e) and wind-based (b, d, f) coincidence comparison methods. On each scatter plot, the red line is the linear fit with intercept set to 0, and the black line is the one-to-one line.**



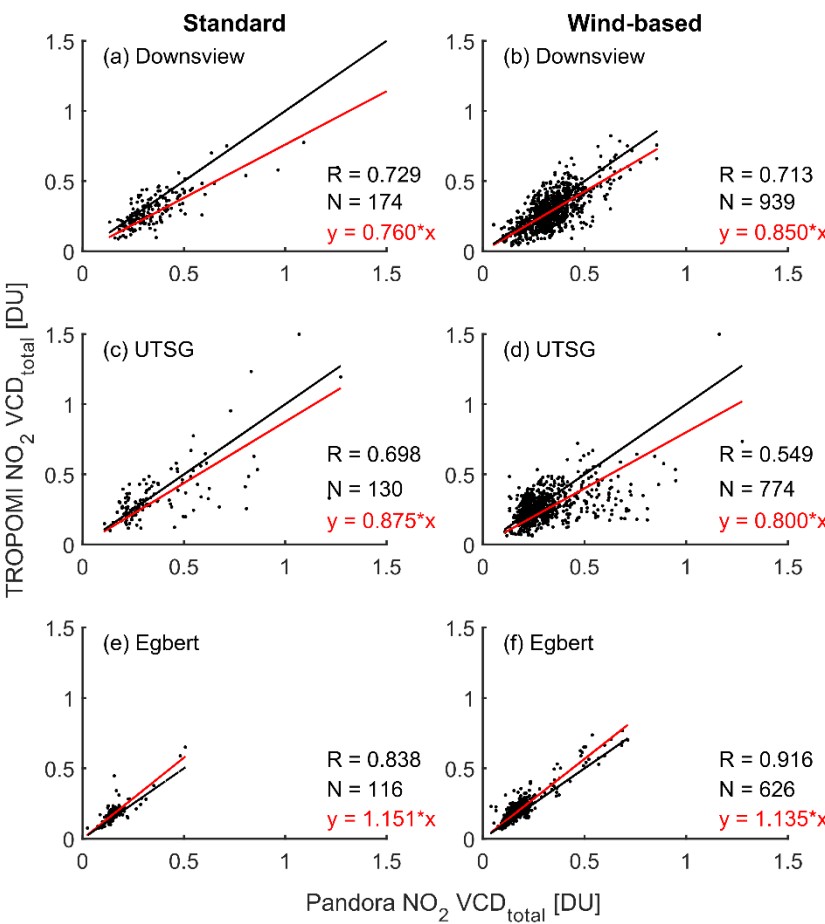

**Figure 5. TROPOMI (ECCC) vs. Pandora NO₂ VCD_total measured at Downsview, UTSG, and Egbert, using the standard (a, c, e) and wind-based (b, d, f) methods. On each scatter plot, the red line is the linear fit with intercept set to 0, and the black line is the one-to-one line.**





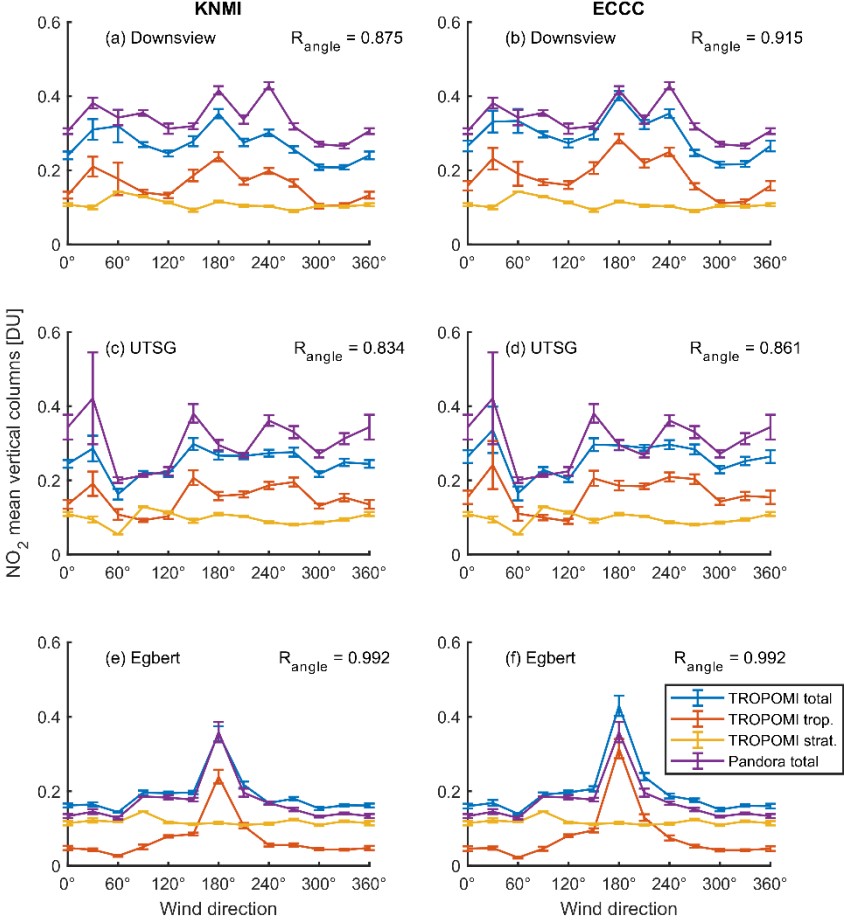

**Figure 6. TROPOMI and Pandora coincident measurements from three sites, binned by wind direction. TROPOMI data in (a), (c), and (e) are KNMI products; (b), (d), and (f) are ECCC products. Blue, red, and yellow lines are TROPOMI total, tropospheric, and stratospheric columns, respectively. Purple lines are Pandora total columns. Error bars are the standard error of the mean. The correlation coefficient between TROPOMI (blue line) and Pandora total column $NO_2$ (purple line) is shown in each panel.**





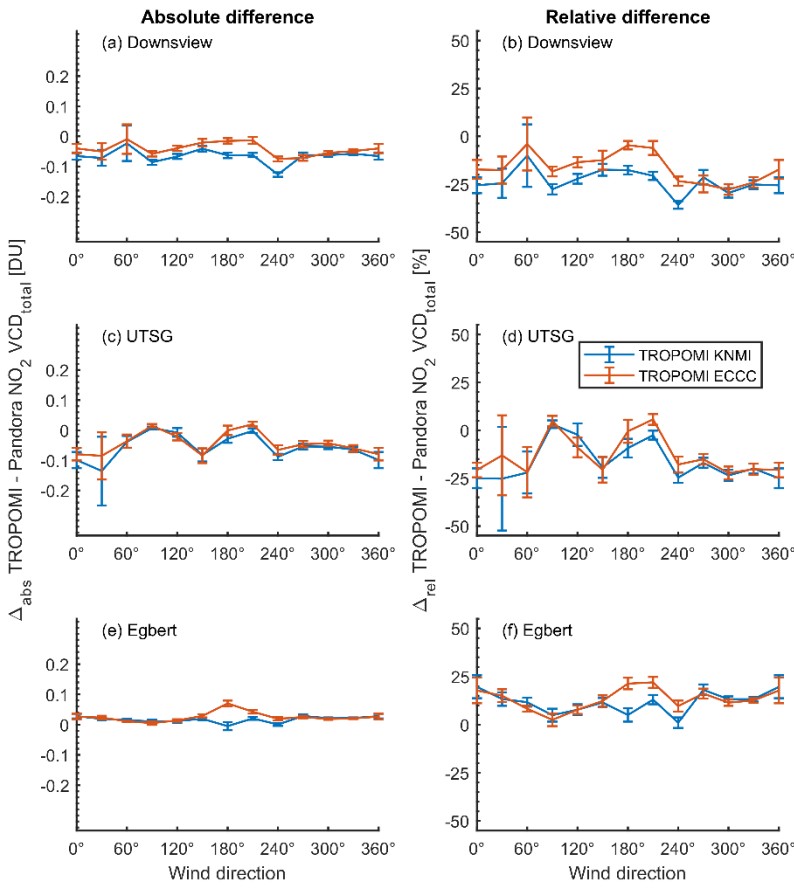

**Figure 7. The absolute and relative differences between TROPOMI and Pandora coincident measurements from three sites, binned by wind direction. TROPOMI and Pandora absolute differences are shown in (a), (c), and (e); their relative differences are shown in (b), (d), and (f). Blue lines are differences calculated using TROPOMI KNMI data products, red lines are their counterparts using ECCC data products. Error bars are the standard error of the mean.**





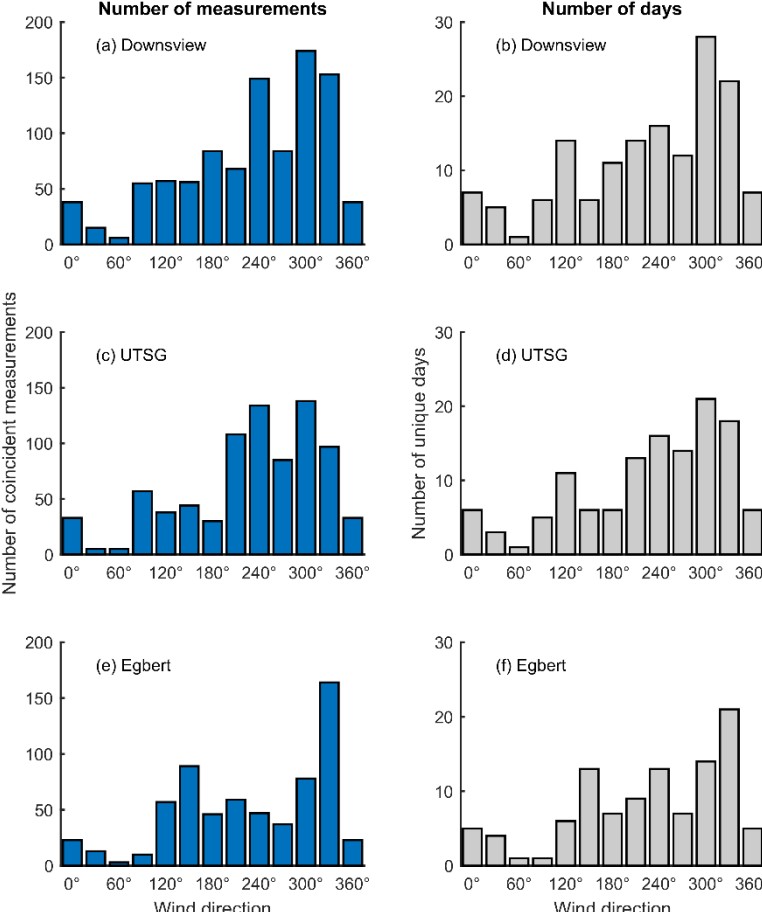

**Figure 8. The number of coincidences of TROPOMI and Pandora measurements and number of unique days for each wind-bin for the data shown in Figures 6 and 7.**

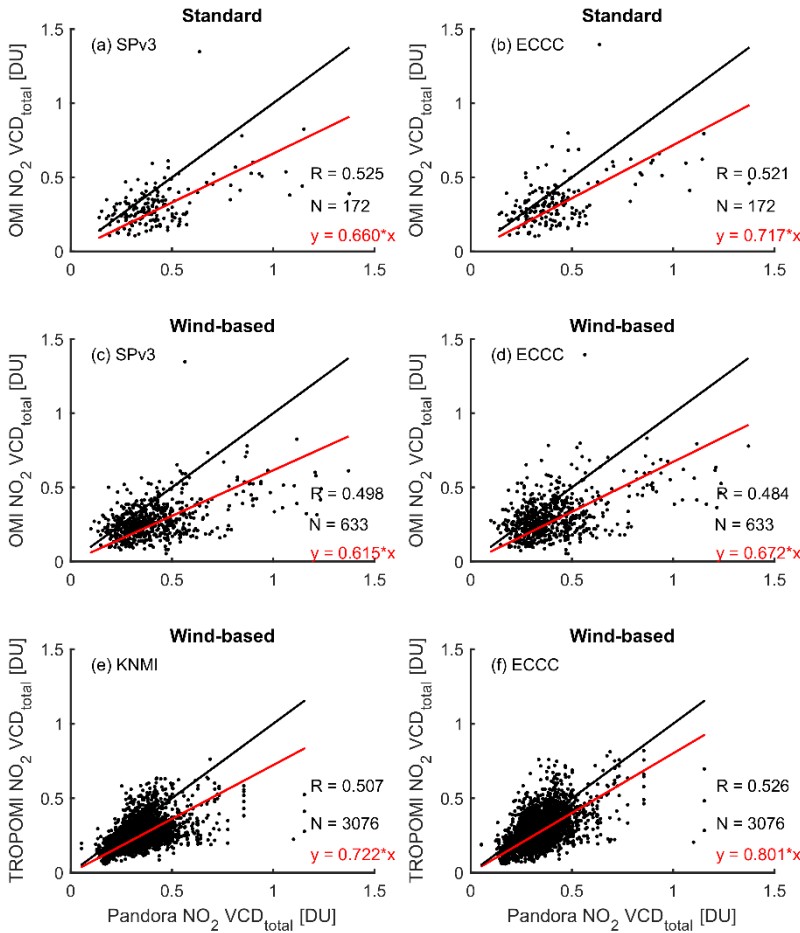

**Figure 9. OMI and TROPOMI vs. Pandora no. 104 (Downsview) NO₂ total columns, using (a, b) the standard coincidence comparison method for OMI SPv3 and OMI ECCC, respectively; (c, d) wind-based method for OMI SPv3 and OMI ECCC, respectively. (e) and (f) are results using the wind-based method for TROPOMI KNMI and ECCC, respectively, with extended $y_{rotate}$ range. On each scatter plot, the red line is the linear fit with intercept set to 0, and the black line is the one-to-one line.**





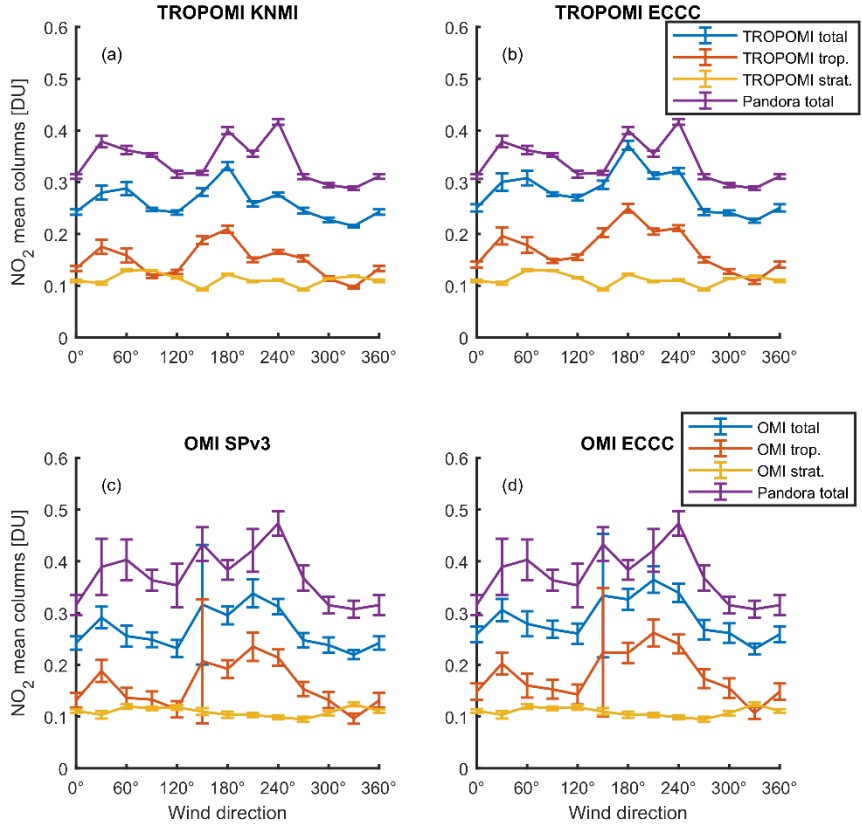

**Figure 10. TROPOMI, OMI, and coincident Pandora measurements from Downsview, binned by wind direction. TROPOMI data in (a) are from KNMI products and (b) are from ECCC products (2018). OMI data in (c) are from NASA SPv3 products and (d) are from ECCC products (2015-2018). Blue, red, and yellow lines are TROPOMI or OMI total, tropospheric, and stratospheric columns. Purple lines are Pandora total columns. Error bars are the standard error of the mean.**



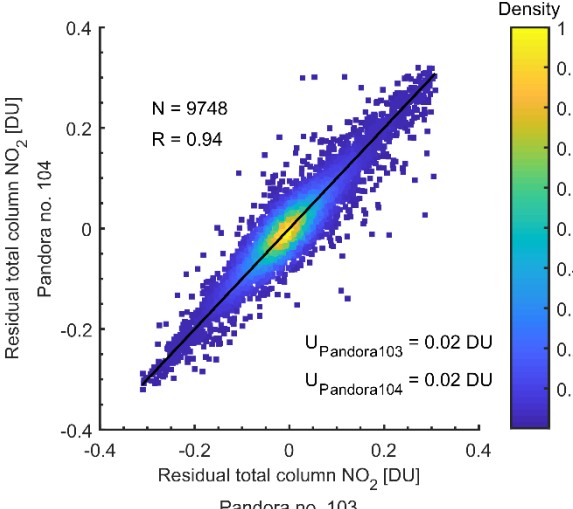

**Figure 11. Scatter plot of residual total column NO₂ measured by Pandora nos. 103 and 104 (2017 December to 2019 June), colour-coded by the normalized density of the points. The black line is the one-to-one line.**





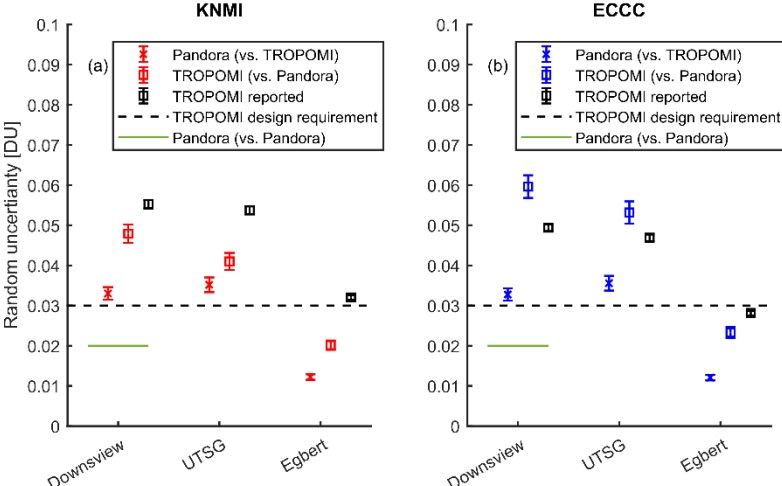

**Figure 12. Statistical uncertainty estimations for TROPOMI and Pandora total column NO₂, using their coincident measurements paired by wind-based methods. (a) TROPOMI KNMI vs. Pandoras at three sites (site names are on the x-axis), the estimated statistical random uncertainties are shown in red with estimated errors. Black squares represent the mean of reported uncertainties for TROPOMI KNMI NO₂ data, with error bars representing the uncertainty of the mean. (b) TROPOMI ECCC vs. Pandoras at three sites (x-axis), the estimated statistical random uncertainties are shown in blue with estimated errors. Black squares represent the mean of reported uncertainties for TROPOMI ECCC NO₂ data, with error bars representing the uncertainty of the mean. The black dashed line is the TROPOMI design requirement for precision, while the green line is the Pandora instrument precision estimated independently (statistical estimation using co-located Pandoras at Downsview).**





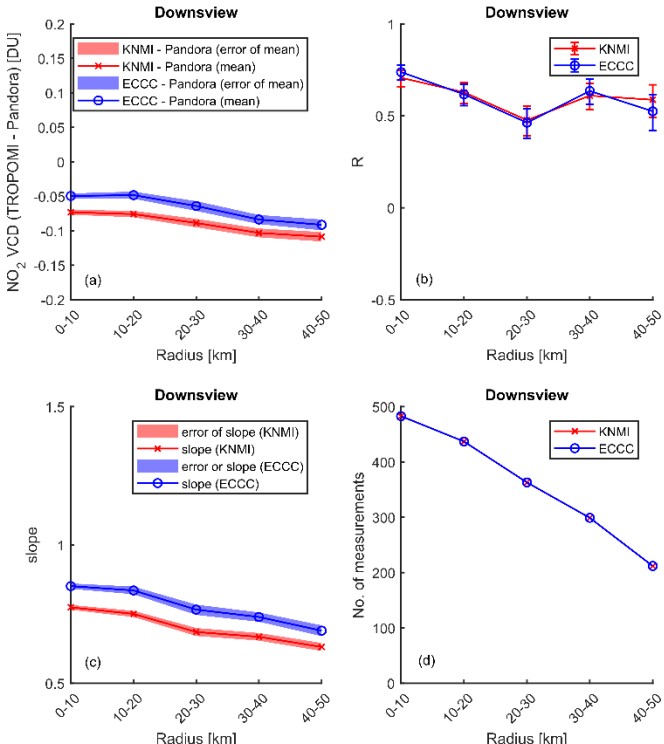

**Figure B1. Sensitivity test for Pandora no. 104 at Downsview. For data within each radius bin, (a) shows the mean difference between TROPOMI and Pandora NO₂ VCD$_{total}$, (b) shows the correlation coefficients, (c) shows the slope (zero offset linear fit), and (d) shows the number of coincident measurements. KNMI data are shown in red, ECCC data are shown in blue. The (a) symmetric standard error of mean and (c) error of slope are shown by colour-coded envelopes, indicated by the legends. The asymmetric error of the correlation coefficients is shown by error bars in (b and d).**





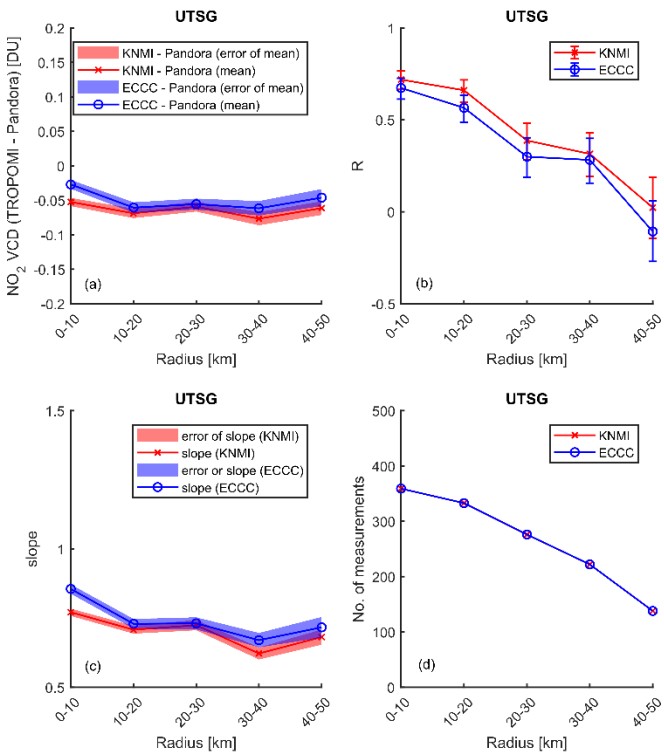

**Figure B2. Sensitivity test for Pandora no. 145 at UTSG. Descriptions of legend in Fig. B1.**





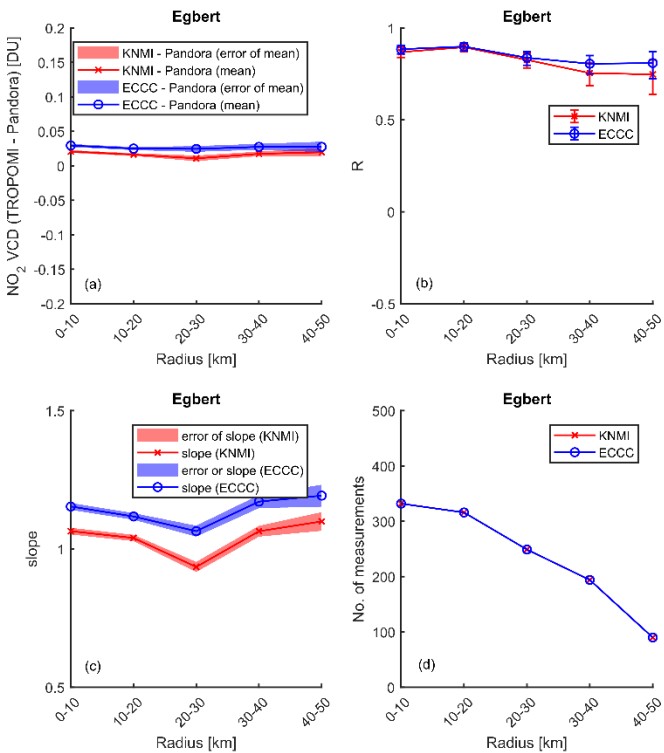

**Figure B3.** Sensitivity test for Pandora no. 108 at Egbert. Descriptions of legend in Fig. B1.





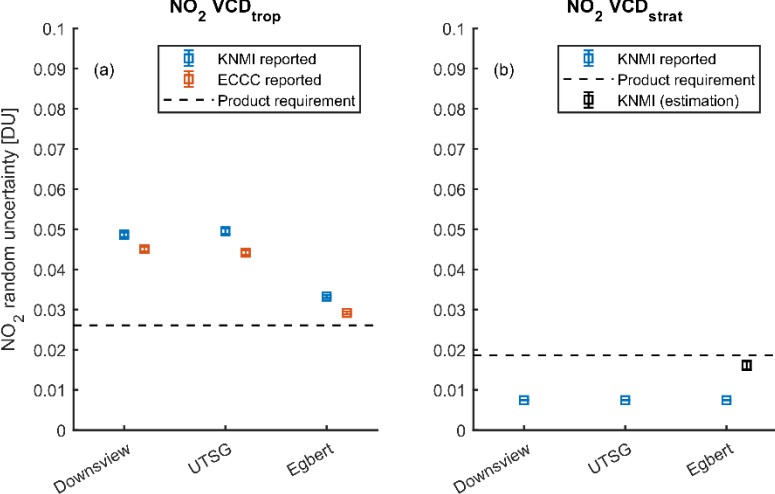

**Figure C1. TROPOMI reported random uncertainties of (a) tropospheric and (b) stratospheric NO₂ columns. Blue squares are KNMI reported random uncertainties, with error bars from the uncertainty of the mean. Red squares are ECCC reported random uncertainties. The black dash lines are the design requirements. The black square represents the estimated uncertainty of KNMI stratospheric data.**





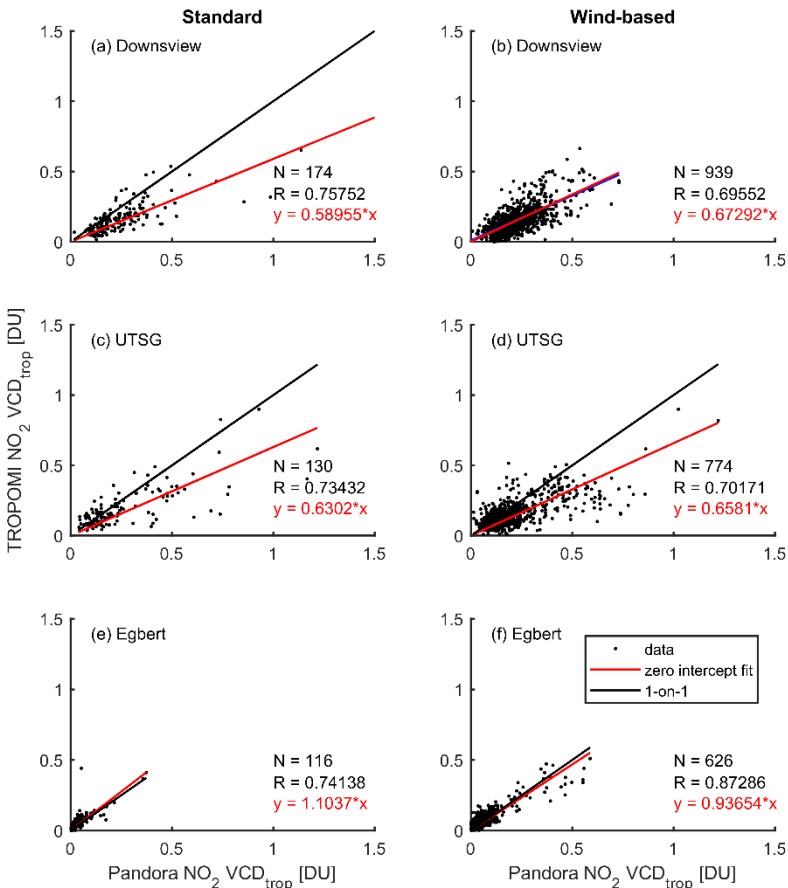

**Figure C2. TROPOMI KNMI NO₂ VCD_trop vs. Pandora NO₂ VCD_trop.**



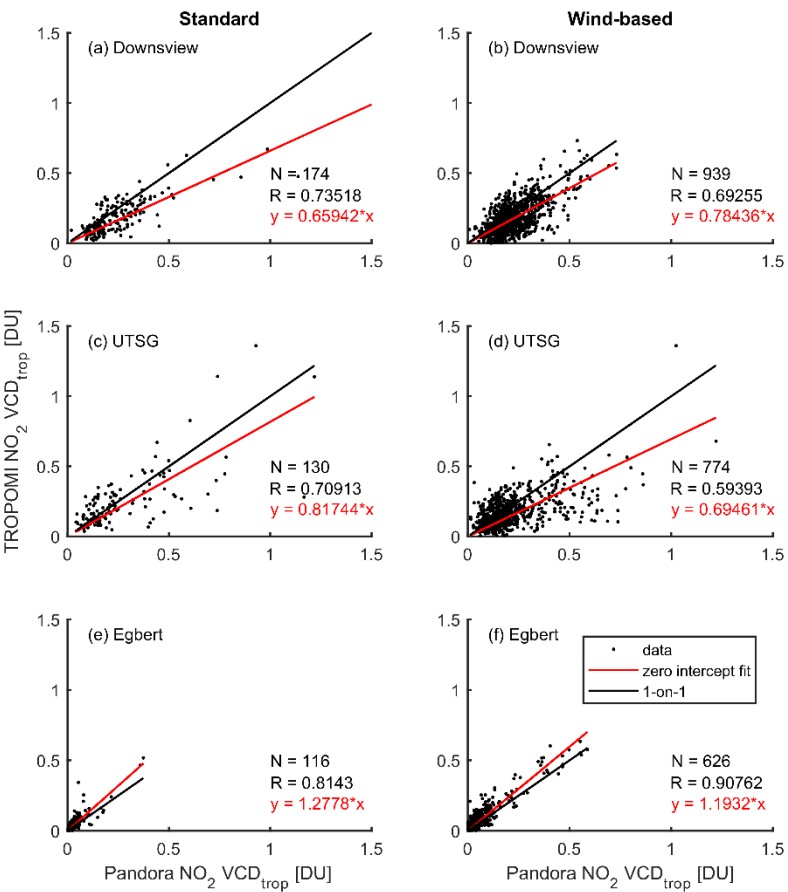

**Figure C3. TROPOMI ECCC NO₂ VCD_trop vs. Pandora NO₂ VCD_trop.**

