# Peer review of "Assessment of the quality of TROPOMI high-spatial-resolution NO2 data products in the Greater Toronto Area"

_Atmospheric Measurement Techniques, 2019_

## Referee Comment (RC1) · Anonymous Referee #1 · 10 Jan 2020

The manuscript present the comparison between TROPOMI and Pandora NO2 retrievals over 3 Canadian sites. The validation is really accurate especially concerning the uncertainty analysis and presents new wind-based approach that provides insight of the relative effect of local emission and transport. The manuscript can be published after addressing the following minor comments.

Specific comments

I have an issue with the title. I think it should include that the assessment of TROPOMI NO2 data in the GTA. Now it gives the impression that the assessment is exhaustive. Maybe you could add "in the Greater Toronto Area" at the end or something similar

Section 2.1.2. Does ECCC NO2 Modis-based albedo include geometry-dependent

information?

Section 3.1 Did you analyse how the agreement change in the standard approach if you change the radius (spatial averaging) or the time range? And did you find any dependence on the pixel size (it should be less important than for OMI, since the increase from the center to the side of the swath is much smaller).

Abstract: You might want to add some number e.g. absolute or relative difference or correlation between Pandora and Tropomi NO2 in the abstract. Also for the improvement in the agreement using high resolution model AMFs it could be useful to quantify this improvement here in the abstract.

P2 L21 You write that Pandora NO2 VCDs have been validated through [...] satellite validations That sounds inaccurate. Maybe you mean "have been used in..."?

P2 L23 You can add these TROPOMI validation paper using Pandora data also here: Herman et al 2019 Ialongo, I. et al, 2019.

P3 L32 Maybe you can mention that the resolution decreased to 3.5 × 5.5 km since 6 August 2019.

P4 L20 TROPOMI file includes a QA quality flag that is recommended to be used for flagging with QA>0.75 for clear sky. You say here that you use cf<0.3 but later on you say that you use the quality flag which already include the cloud screening: can you clarify?

P8 L18 In previous -> Previously

P10 L20 You find positive bias at Egbert: can you speculate on the reasons? Stratospheric overestimation perhaps (see e.g. Wang et al 2019 AMTD)?

P14 L9 TROMPOMI -> TROPOMI

Figure 1. It could be useful to add (perhaps in the appendix or supplement as well) a map like Fig. 1 including OMI data in order to visualise the differences in the mapping

capability of the two instruments.

Figure 8. Could be this go to appendix or supplement? (the paper is quite figure heavy anyway)

---

## Referee Comment (RC2) · Anonymous Referee #2 · 11 Feb 2020

This paper presents an evaluation of the KNMI standard and ECCC TROPOMI NO2 data products based on comparisons to Pandora NO2 measurements at three sites located in the Greater Toronto Area. In addition to the traditional pairing of ground-based measurements with satellite observations closest in time and space, a wind-based validation technique making use of the TROPOMI pixels located upwind and downwind from the Pandora sites is also presented. With this technique, the number of coincident measurements can be significantly increased, allowing to reveal detailed spatial patterns of local and transported NO2 emissions. This study also showed that the TROPOMI ECCC NO2 research data product based on AMFs calculated from high-resolution regional air quality forecast model is in general in better agreement with Pandora measurements compared to the standard product.

This paper is very well written and presents interesting results which fit well with the scope of AMT. I recommend the manuscript for final publication after addressing the following specific comments:

Title: The title should reflect the fact that this study is limited to three sites located in the Toronto area and is therefore not a global assessment of TROPOMI NO2 data products.

Page 2, lines 21-24: It is written that the Pandora direct-sun NO2 VCD products have been validated through...satellite validations. This is a bit weird and the sentence should be rephrased.

Page 3, lines 5-7: The formulation is also a bit odd here. Suggestion: 'These AMFs were found to lead to a better agreement with aircraft....'. Also, have those new AMFs been also validated in other locations than the Athabasca Oil Sands Region, which corresponds to very specific conditions?

Page 5, lines 7-9: Why using two different albedo products for areas with and without snow ?

Page 6, lines 25-27: The three Pandora instruments have an alternate observation schedule (direct-sun/zenith-sky/multi-axis). Was there any attempt to use these three viewing modes synergistically, e.g. using the multi-axis tropospheric NO2 columns to check those retrieved from the direct-sun mode or to evaluate the stratospheric columns based on the zenith-sky observations instead of using the TROPOMI strato-spheric columns to correct for the contribution of the stratosphere ?

Page 8, lines 7-8: More details should be given here about the QA/QC selection criteria applied to the Pandora direct-sun NO2 total column data used in this study.

Page 12, lines 32-33: the 240° peak is more influenced by some near-local NO2 sources. Do you have any idea about those potential sources ? If yes, you should add them here.

Page 17, lines 1-4: Using a high-resolution regional air quality forecast model in the TROPOMI AMF calculation improves the agreement with Pandora data in urban conditions but not at a rural site like Egbert. What could be the reason for that ? Is it an indication that the GEM-MACH model does not perform well in background conditions ? Could it be related to the albedo product used in the retrieval ? Maybe this point should be further commented in the Conclusion section ?

Technical corrections:

Page 1, line 23: in order to avoid the repetition of 'use', I would replace '(used in the air mass factor calculation).' by 'in the air mass factor calculation'.

Page 1, line 31: same remark as above ('Using this larger number...' -> 'With this larger number....').

Page 2, line 25: 'Funded by the European Space Agency (ESA),...'

Page 3, line 28: '...near full Earth' surface coverage...'

Page 14, line 9: 'TROMPOMI' -> 'TROPOMI'

---

## Author Comment (AC1) · 6 Mar 2020

**Response to Referee #1:**

We thank referee #1 for their helpful comments. Our responses are given below in black with the referee's comments in blue. The new/revised text in the modified manuscript is given in red (italicized).

Specific comments

I have an issue with the title. I think it should include that the assessment of TROPOMI NO2 data in the GTA. Now it gives the impression that the assessment is exhaustive. Maybe you could add "in the Greater Toronto Area" at the end or something similar

Done. The title has been modified as requested.

*Assessment of the quality of TROPOMI high-spatial-resolution NO$_2$ data products in the Greater Toronto Area*

Section 2.1.2. Does ECCC NO2 Modis-based albedo include geometry-dependent information?

No. The product used is a simple Lambertian albedo (MODIS product MCD43C3), and therefore, is not dependent on geometry (analogous to the albedo product used in the KNMI TROPOMI AMF calculation). The TROPOMI ATBD albedo (https://sentinel.esa.int/documents/247904/2476257/Sentinel-5P-TROPOMI-ATBD-NO2-data-products) estimates using this approach, as opposed to a BRDF, which leads to an error of roughly 5% or less (page 35), citing Zhou et al., 2009. The sentence is modified in the text.

*Improved albedo inputs were created using averaged monthly albedo for areas without snow cover and a climatology for snow-covered areas using the MODIS MCD43C3 data product (Schaaf et al., 2002) by only considering grid-boxes that were 100% snow-free or 100% snow-covered. The choice of which to use, snow-free or snow-covered, is determined using the IMS snow product.*

Section 3.1 Did you analyse how the agreement change in the standard approach if you change the radius (spatial averaging) or the time range? And did you find any dependence on the pixel size (it should be less important than for OMI, since the increase from the center to the side of the swath is much smaller).

Yes, we tested the standard approach with various radius and time range criteria. The current criteria (d < 20 km, t < ±10 min) selected are found to give a good balance between the number and quality of coincident measurements.

For TROPOMI data, we did not filter the pixels by their footprint. For the GTA area (before Aug. 6$^{th}$, 2019), we found the smallest central pixel has a footprint of 26 km$^2$ (approximately 7 km × 3.7 km), whereas the largest edge pixel has a footprint of only 100 km$^2$ (approximately 7 km × 14 km). Thus, even TROPOMI's large pixel has a spatial resolution better than OMI's "small pixel". To reveal the potential dependence on TROPOMI's pixel size as suggested by the referee, we need more ground-based and satellite coincident data (i.e., three or more years of data might be sufficient). With the current one-year data, it is difficult to make a solid conclusion.

Abstract: You might want to add some number e.g. absolute or relative difference or correlation between Pandora and Tropomi NO2 in the abstract. Also for the improvement in the agreement using high resolution model AMFs it could be useful to quantify this improvement here in the abstract.

Done.

*It is found that these current TROPOMI tropospheric NO$_2$ data products (standard and ECCC) met the TROPOMI design bias requirement (<10 %).*

*The Pandora instruments are found to have sufficient precision (<0.02 DU) to perform TROPOMI validation work.*

*The TROPOMI ECCC NO$_2$ research data product shows improved agreement with Pandora measurements compared to the TROPOMI standard tropospheric NO$_2$ data product (e.g., lower multiplicative bias at the suburban and urban sites by about 10 %), demonstrating benefits from the high-resolution regional air quality forecast model.*

P2 L21 You write that Pandora NO2 VCDs have been validated through […] satellite validations That sounds inaccurate. Maybe you mean "have been used in…"?

Done.

*The Pandora direct-sun NO$_2$ VCD$_{total}$ products have been validated through many field campaigns (Flynn et al., 2014; Lamsal et al., 2017; Martins et al., 2016; Piters et al., 2012; Reed et al., 2015), ground-based comparisons (Herman et al., 2009; Wang et al., 2010), and used in satellite validations ( Griffin et al., 2019; Herman et al., 2019; Ialongo et al., 2016, 2019; Lamsal et al., 2014).*

P2 L23 You can add these TROPOMI validation paper using Pandora data also here: Herman et al 2019 Ialongo, I. et al, 2019.

New citations are included.

P3 L32 Maybe you can mention that the resolution decreased to 3.5  5.5 km since 6 August 2019.

This information has been included.

*The instrument has a high spatial resolution of 7 km × 3.5 km (along-track × across-track) at nadir for bands 2-6 (UVN module) (Eskes et al., 2019)* *(note that since 6 August 2019, the resolution improved to 5.5 km × 3.5 km).*

P4 L20 TROPOMI file includes a QA quality flag that is recommended to be used for flagging with QA>0.75 for clear sky. You say here that you use cf<0.3 but later on you say that you use the quality flag which already include the cloud screening: can you clarify?

This extra cf<0.3 filter is used to ensure the comparison between OMI and TROPOMI is straightforward, i.e., we used a cf<0.3 filter for OMI data. The explanation has been included in P8.

*Note that the TROPOMI quality assurance value filter (qa_value > 0.75) removes cloud-covered scenes with cloud radiance fraction > 0.5. In this study, to make a straightforward comparison with OMI, an additional cloud fraction filter is used (cloud fraction <= 0.3) for TROPOMI data.*

P8 L18 In previous -> Previously

Done.

P10 L20 You find positive bias at Egbert: can you speculate on the reasons? Stratospheric overestimation perhaps (see e.g. Wang et al 2019 AMTD)?

We agree with the referee that stratospheric overestimation in the TROPOMI stratospheric columns could be a reason for this positive bias at Egbert. Currently, Pandora only has a total column $NO_2$ data

product. In the future, Pandora tropospheric and stratospheric column products (e.g., products from zenith-sky and multi-axis measurements) can be used to further this investigation.

*The positive bias at Egbert might be due to TROPOMI overestimating stratospheric $NO_2$ (e.g., Wang et al., 2019).*

Done.

Figure 1. It could be useful to add (perhaps in the appendix or supplement as well) a map like Fig. 1 including OMI data in order to visualise the differences in the mapping capability of the two instruments.

The pixel-averaging plot for OMI is made with its 2015-2018 data (see Fig. R1 in below). The averaging radius is selected to be the same as Fig. 1 (i.e., 7 km). In general, the spatial distribution of high-density $NO_2$ over the GTA area is consistent with the results in Fig. 1, which use TROPOMI data. Although some small-scale features are not identical, these differences might be due to their different averaging periods (i.e., three years for OMI, but one year for TROPOMI). To fully reveal the mapping capability of the two instruments, we shall wait for another two or more years of coincident measurements. Also, as suggested by the referee that this paper is already figure-heavy, we decided not to include this extra map in the paper.

[Figure]

**Figure R1. OMI SPv3 NO$_2$ tropospheric columns smoothed by pixel averaging (2015 to 2018) (© Google Maps).**

Figure 8. Could be this go to appendix or supplement? (the paper is quite figure heavy anyway)

Done. Figure 8 has been moved to Appendix A.

[revised manuscript text omitted]

---

## Author Comment (AC2) · 6 Mar 2020

**Response to Referee #2:**

We thank referee #2 for their helpful comments. Our responses are given below in black with the referee's comments in blue. The new/revised text in the modified manuscript is given in red (italicized).

Referee #2:

Title: The title should reflect the fact that this study is limited to three sites located in the Toronto area and is therefore not a global assessment of TROPOMI NO2 data products.

Done. The title has been modified as requested.
*Assessment of the quality of TROPOMI high-spatial-resolution NO$_2$ data products in the Greater Toronto Area*

Page 2, lines 21-24: It is written that the Pandora direct-sun NO2 VCD products have been validated through...satellite validations. This is a bit weird and the sentence should be rephrased.

Done.
*The Pandora direct-sun NO$_2$ VCD$_{total}$ products have been validated through many field campaigns (Flynn et al., 2014; Lamsal et al., 2017; Martins et al., 2016; Piters et al., 2012; Reed et al., 2015), ground-based comparisons (Herman et al., 2009; Wang et al., 2010), and used in satellite validations ( Griffin et al., 2019; Herman et al., 2019; Ialongo et al., 2016, 2019; Lamsal et al., 2014).*

Page 3, lines 5-7: The formulation is also a bit odd here. Suggestion: 'These AMFs were found to lead to a better agreement with aircraft … .'. Also, have those new AMFs been also validated in other locations than the Athabasca Oil Sands Region, which corresponds to very specific conditions?

Done. The validation work for the Athabasca Oil Sands Region was the first implementation of the ECCC recalculated AMFs to TROPOMI. The current paper is the second research paper to further the validation work. The oil sands region is considered a very special case as it has high-level emissions limited in a

relative small area (compare to GTA). The Pandora sites in the GTA areas provide a few different environmental conditions to further validate the ECCC-recalculated AMFs for TROPOMI.

*These AMFs were found to* *lead to a* *better agreement with aircraft and ground-based measurements in the Athabasca Oil Sands Region (AOSR) (Griffin et al., 2019) than the standard TROPOMI tropospheric $NO_2$ (referred to as KNMI $NO_2$).*

Page 5, lines 7-9: Why using two different albedo products for areas with and without snow?

The albedo will change considerably from one day to another if snow appears or disappears.  To account for this, both snow-covered and snow-free albedo databases were created from the MODIS albedo data product.  The choice of albedo is made based on whether the IMS snow product indicates the presence of snow on that day. This is clarified by adjusting the text as follows:

*Improved albedo inputs were created using averaged monthly albedo for areas without snow cover and a climatology for snow-covered areas* *using the MODIS MCD43C3 data product (Schaaf et al., 2002) by only considering grid-boxes that were 100% snow-free or 100% snow-covered.  The choice of which to use, snow-free or snow-covered, is determined using the IMS snow product.*

Page 6, lines 25-27: The three Pandora instruments have an alternate observation schedule (direct sun/zenith-sky/multi-axis). Was there any attempt to use these three viewing modes synergistically, e.g. using the multi-axis tropospheric NO2 columns to check those retrieved from the direct-sun mode or to evaluate the stratospheric columns based on the zenith-sky observations instead of using the TROPOMI stratospheric columns to correct for the contribution of the stratosphere ?

The idea suggested by the referee is very interesting and is part of our algorithm development plan, i.e., synergistically use the data products from alternate modes. Currently, PGN has developed a new O2-O2 ratio algorithm (Cede, 2019) to retrieve tropospheric $NO_2$ columns from Pandora's multi-axis measurements. The ECCC Pandora program also developed a Pandora zenith-sky data product (Zhao et al., 2019). The multi-modes capability of Pandora instruments provides us with a unique opportunity to derive various data products that suit different needs from one single instrument.

Page 8, lines 7-8: More details should be given here about the QA/QC selection criteria applied to the Pandora direct-sun NO2 total column data used in this study.

Done. The Pandora direct-sun $NO_2$ quality flag information is included.

*Pandora direct-sun $NO_2$ total column data of assured high-quality (L2 data quality flag = 0) are used in the validation (Cede, 2019).*

Page 12, lines 32-33: the 240 peak is more influence by some near-local NO2 sources. Do you have any idea about those potential sources? If yes, you should add them here.

Done. The information has been included.

*Thus, the 240° peak is more influenced by some near-local $NO_2$ source (e.g., nearby heavy traffic roads).*

Page 17, lines 1-4: Using a high-resolution regional air quality forecast model in the TROPOMI AMF calculation improves the agreement with Pandora data in urban conditions but not at a rural site like Egbert. What could be the reason for that. Is it an indication that the GEM-MACH model does not perform well in background conditions? Could it be related to the albedo product used in the retrieval? Maybe this point should be further commented in the Conclusion section?

We thank referee for this important question. There are several factors that could contribute to this difference found in rural and urban sites. However, this is not likely an issue with the albedo used in the ECCC AMF calculation. The MODIS albedo used, when smoothed, is very similar to the coarse albedo used by KNMI (see, e.g., McLinden et al., 2012, Figure 3). The different performance in the rural site could be related to the GEM-MACH model, but overall, obtaining an accuracy from satellite of better than 10-20% is challenging as there are several potential sources of bias that are of this magnitude. For example, using a Lambertian albedo as opposed to a BDRF approach may cause a bias of 5%, and using AMFs based on an aerosol-free atmosphere (and assuming it is compensated for in the cloud-fraction) is also a source of bias. Out-of-date emissions can lead to bias in the model profiles. In short, squeezing out this remaining 10-20% is beyond the ability of current algorithms. Thus, unfortunately, it is difficult to draw any solid conclusion.

Technical corrections:

Page 1, line 23: in order to avoid the repetition of 'use', I would replace '(used in the air mass factor calculation).' by 'in the air mass factor calculation'.

Done.

Page 1, line 31: same remark as above ('Using this larger number…' -> 'With this larger number….').

Done.

[revised manuscript text omitted]